# Regional patterns of human cortex development correlate with underlying neurobiology

Leon D. Lotter [1,2,3] ✉, Amin Saberi [1,2,4], Justine Y. Hansen [5], Bratislav Misic [5], Casey Paquola[1], Gareth J. Barker [6], Arun L. W. Bokde [7], Sylvane Desrivières [8], Herta Flor [9,10], Antoine Grigis[11], Hugh Garavan [12], Penny Gowland [13], Andreas Heinz [14], Rüdiger Brühl [15], Jean-Luc Martinot [16], Marie-Laure Paillère[16,17], Eric Artiges [16,18], Dimitri Papadopoulos Orfanos [11], Tomáš Paus [19,20], Luise Poustka[21], Sarah Hohmann [22], Juliane H. Fröhner [23], Michael N. Smolka [23], Nilakshi Vaidya [24], Henrik Walter [14], Robert Whelan [25], Gunter Schumann [24,26], IMAGEN Consortium*, Frauke Nees [9,22,27], Tobias Banaschewski [22,28], Simon B. Eickhoff[1,2] & Juergen Dukart [1,2] ✉

Human brain morphology undergoes complex changes over the lifespan. Despite recent progress in tracking brain development via normative models, current knowledge of underlying biological mechanisms is highly limited. We demonstrate that human cortical thickness development and aging trajectories unfold along patterns of molecular and cellular brain organization, traceable from population-level to individual developmental trajectories. During childhood and adolescence, cortex-wide spatial distributions of dopaminergic receptors, inhibitory neurons, glial cell populations, and brain-metabolic features explain up to 50% of the variance associated with a lifespan model of regional cortical thickness trajectories. In contrast, modeled cortical thickness change patterns during adulthood are best explained by cholinergic and glutamatergic neurotransmitter receptor and transporter distributions. These relationships are supported by developmental gene expression trajectories and translate to individual longitudinal data from over 8000 adolescents, explaining up to 59% of developmental change at cohort- and 18% at single-subject level. Integrating neurobiological brain atlases with normative modeling and population neuroimaging provides a biologically meaningful path to understand brain development and aging in living humans.

The human cerebral cortex develops in complex patterns[1–3], giving rise to our cognitive abilities[4,5]. Biologically, these morphological changes are likely driven by developmental processes originating from different organizational levels. Microstructural reorganization, e.g.,

neuronal and glial restructuring, synaptic remodeling ("pruning"), as well as pericortical myelination, has been discussed as the main driver of cortical thickness (CT) development[6–9]. The neuronal component mainly consists in remodeling of dendritic arbor, with human

A full list of affiliations appears at the end of the paper. *A list of authors and their affiliations appears at the end of the paper. ✉e-mail: l.lotter@fz-juelich.de; leondlotter@gmail.com; juergen.dukart@gmail.com

postmortem evidence for increases of synaptic and dendrite density into childhood and early adolescence, followed by gradual decreases during adolescence[10–12] that might extend even into adulthood[13]. As microscale developments on the synapse level alone are unlikely to explain macroscale CT changes, childhood and adolescence neuronal remodeling is likely to be accompanied by a changes of cortical glial cells[6,14], in line with, e.g., microglia potentially playing an active role in developmental synaptic remodeling[7]. In contrast, pericortical myelination is thought to specifically influence magnetic resonance imaging (MRI)-based CT through myelin-dependent changes in tissue contrasts, which might result in apparent cortical thinning[6,8].

Given the multitude of neurobiological mechanisms that likely shape cortex morphology over the lifetime, it is to assume that CT change patterns at any given developmental period result from several interacting biological factors jointly influencing cortical microstructure as outlined above. For example, concerted developments across cortical cell populations could be mediated via specific neurotransmitter effects projected from deeper brain regions, as was indicated in early non-human animal studies for, e.g., glutamatergic[15] and serotonergic[16] effects of thalamocortical projections on motor and somatosensory cortices as well as for dopaminergic effects of mesocortical projections on the prefrontal cortex[17]. Relatedly, neurotransmitter receptors likely play regulatory roles in cortical development as evidenced, for instance, by the effects of in-utero cocaine exposure on cortical macrostructure[18], thought to be caused by a disruption D1 and D2 dopaminergic receptor influences during cortex development[19]. Unfortunately, as today's neuroimaging tools do not suffice to study human cellular neurobiology in detail, we have to rely on scarce human postmortem and non-human animal data. Conversely, while structures and processes on the molecular level are partly accessible in humans with nuclear imaging, here, the exposure to radioactivity practically forbids application in typical developing children and adolescents, limiting its use to study neurodevelopment. While the study of neurobiological mechanisms underlying human brain development suffers from these practical challenges, considerable progress has been made in mapping the development and aging of human brain macrostructure, with large-scale normative models[1,3,20] providing new insights in both population-level typical and individual atypical neurodevelopment[21,22]. Similarly, our understanding of general brain organization was significantly advanced by the availability of modern in vivo nuclear imaging atlases[23,24], explaining typical brain organization and disordered brain structure to greater extents as compared to MRI-based brain structural and functional metrics[25,26].

Neural cell populations and molecular-level tissue structures and processes – their neuroimaging-based correlates hereafter collectively referred to as "neurobiological markers" – are not uniformly distributed across the cortex, but show distinct spatial distributions[27–30]. Similarly, CT development and aging trajectories vary by cortex region[3,11], resulting in distinct spatial change patterns associated with any given developmental period. We assume that these CT change patterns are not random but reflect neurobiological processes that causally influence CT changes over time. To elaborate our rationale (Supplementary Text S1.1 for a more detailed account), let $X$ be a neurobiological entity that exhibits a non-uniform distribution across cortical regions, changes with neurodevelopment and aging, and might have direct or indirect downstream effects on CT. While $X$'s spatial distribution might change across the human lifespan, major distribution changes are more likely during childhood and adolescence (and again during aging) than compared to a relatively stable middle adulthood period. If changes in $X$ lead to changes in CT, the spatial distribution of CT changes likely resembles $X$'s steady state, as we would assume regions with higher final density of $X$ to have shown stronger developmental activity. We conclude that an observed "spatial colocalization" – the alignment of spatial patterns between two measured brain metrics – between (i) the distribution of $X$ as measured

during the stable period and (ii) the distribution of CT changes during a given developmental period could have resulted from a developmental or aging process that $X$ is subject to. Notably, it is conceivable that a third process could influence both, $X$ and CT, leading to a correlation between $X$ and CT changes that is non-causal but still implies a neurobiological mechanism influencing both. Applying a similar reasoning, prior spatial colocalization studies have demonstrated that spatial patterns of CT development are correlated with adult distributions of glial cells, pyramidal neurons, and neuronal cell components[31–36], providing the majority of recent evidence for human cortex-developmental mechanisms[6,37]. Of note, all cited studies are based only on bulk-sequencing postmortem gene expression data from the Allen Brain Atlas[38], which may only poorly represent the in vivo expression patterns for many genes[39]. We emphasize that, despite causal assumptions being made on the conceptual side (see above), neither prior nor the current spatial correlation study can actually prove causal relationships between an MRI-observed change pattern and tested neurobiological markers. Relatedly, the specificity of spatial associations is inherently limited by the spatial resolution and noise associated with both correlated components.

Our current knowledge on biological factors that guide typical human CT development is severely limited by practical obstacles. Multimodal neuroimaging-based spatial colocalization approaches can provide a window into specific biological mechanisms, but – to our knowledge – developmental studies until now were limited to postmortem data. Combining these approaches with date's availability of large-scale normative models and in vivo derived molecular brain atlases constitutes the next major step in the imaging-based study of human brain development (Fig. 1). If translated to the level of the individual subject, the approach can serve as the foundation for future neuroimaging-based yet biologically interpretable biomarkers to be tested for their clinical potential[23,40,41].

Following this reasoning, in this work, we explore if and to what extent spatiotemporal patterns of CT change modeled throughout the human lifespan are explained by the spatial distributions of underlying neurobiological properties, and whether associations observed on the population level translate to individuals. We find that cortex-wide distributions of dopaminergic receptors, inhibitory neurons, glial cell populations, and brain-metabolic features can account for up to 50% of the interregional variance arising from CT development during childhood and adolescence. Cholinergic and glutamatergic neurotransmitter systems, however, best explain adult CT change patterns. The observed developmental associations replicate in independent longitudinal data, albite showing strong interindividual variance. Our study provides a foundation and blueprint for future investigations, exploring developmental spatial colocalization analyses as a tool for understanding both typical and atypical human neurodevelopment.

## Results
### Molecular and cellular neurobiological markers
We collected (i) 21 postmortem gene-expression "cellular markers" mapping neuronal and glial cell populations[38,42,43], (ii) 27 in vivo nuclear imaging "molecular markers" of neurotransmitter receptors and transporters, synaptic density, transcriptomic activity[23–25], as well as, to cover further potentially relevant factors, of brain metabolism and immune activity[24,44,45], and (iii) an MRI-derived map of cortical microstructure (T1w/T2w)[25] (Fig. 1A and S1; all derived from independent healthy adult samples: Supplementary Data S1, Supplementary Text S1.2.1 and S1.2.2). In support of our analytical rationale, three neurotransmitter receptors/transporters, for which alternative atlases from different adult age groups were available, showed high stability of spatial patterns during adulthood (Spearman's rho ≥0.68; Text S1.2.3, Fig. S2). The analytic approach taken here establishes associations between temporospatial CT (change) patterns and brain atlases based on the similarity of cortex-wide

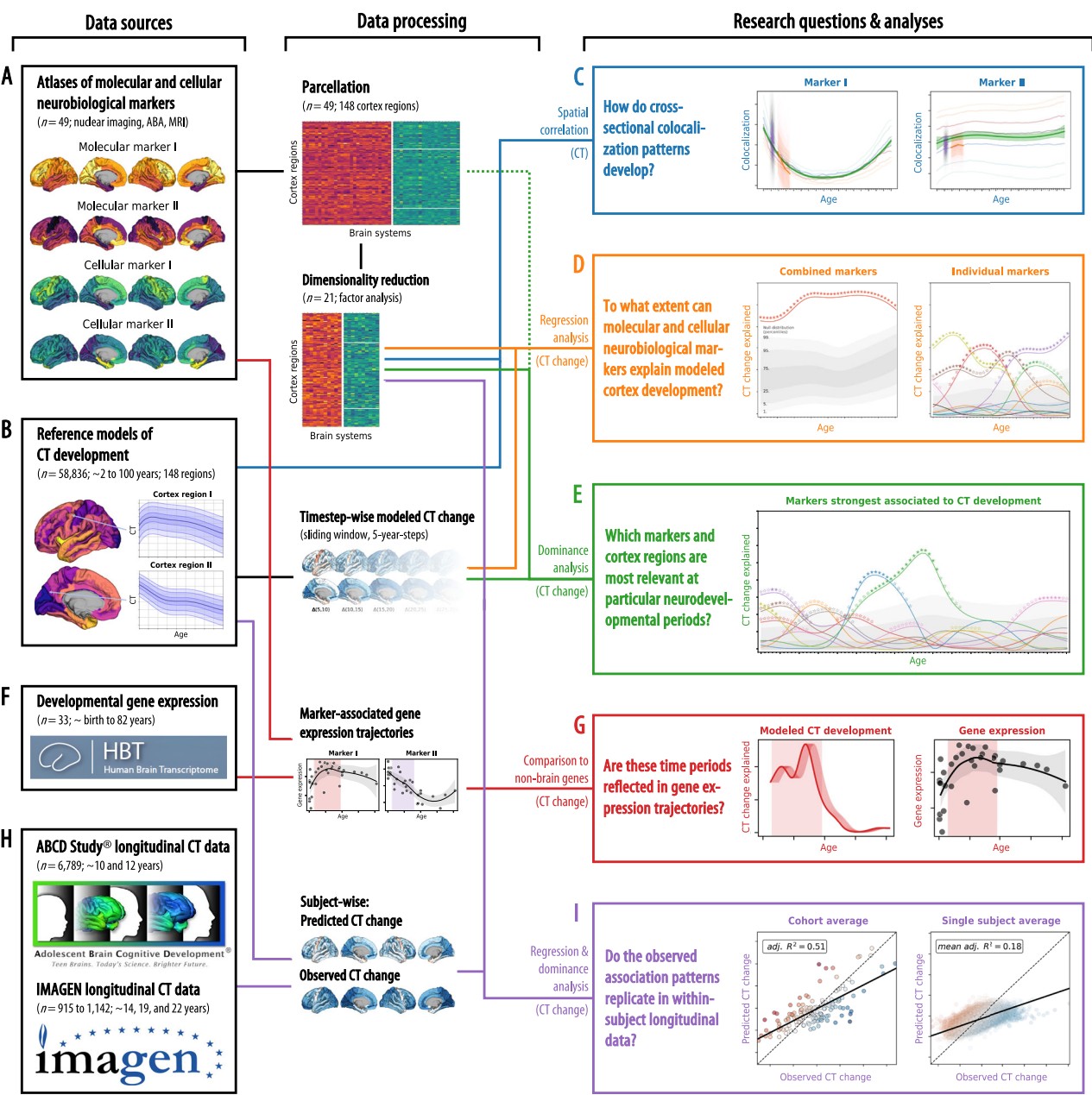

**Fig. 1 | Study overview.** The workflow of the present study, from data sources (left side) to data processing and analysis method (middle) to the research questions and results (right side). **A** A collection of postmortem "cellular" and in vivo "molecular" brain atlases was parcellated and dimensionality reduced. **B** "Modeled" predicted CT data was extracted from a normative model. **C** We calculated the colocalization between neurobiological markers and CT at each point throughout the lifespan (see Fig. 2). **D** We evaluated how combined and individual neurobiological markers could explain lifespan CT change (see Figs. 3 and 4). **E** The strongest associated markers were examined in detail, accounting for shared spatial patterns (see Fig. 5). **F** A developmental gene expression dataset was used to generate trajectories of gene expression associated with each neurobiological marker.

**G** Periods in which CT change was significantly explained were validated in developmental gene expression data (see Fig. 7). **H** Single-subject longitudinal data was extracted from two developmental cohorts. **I** Findings based on the normative model were validated in single-subject data (see Fig. 8). Abbreviations: CT = cortical thickness, ABA = Allen Brain Atlas, MRI = magnetic resonance imaging. Here, data plots are employed for demonstration purposes; for definitions of plot elements, please refer to the individual figures as referrerred to above, similarly, source data are provided in Source Data files of each following figure. ABCD Study®, Teen Brains. Today's Science. Brighter Future.® and the ABCD Study Logos are registered marks of the U.S. Department of Health & Human Services (HHS).

spatial patterns (148 parcels; Destrieux parcellation[46]). Intercorrelation arising from spatial patterns shared between atlases on either cellular or molecular levels (Fig. S3A) was reduced by factor analyses applied independently to the cellular and molecular marker sets after parcellation of the individual maps. For each marker set, all unrotated factors that explained at least 1% of the set's variance were retained, resulting in 10 factor-level nuclear imaging maps (ni1–10) and 10 gene expression cell marker maps (ce1–10).

After promax rotation, these factors explained 90.9% and 86.9% of each marker set's variance, respectively (Fig. S3B, C). We chose the liberal factor-number criterion to balance retaining as much of the spatial information in the neurobiological markers as possible with reducing marker multicollinearity in the following multivariate analyses. Factor solutions were successfully validated against permuted brain maps (Text S1.2.4) and factors were named based on the most closely related original atlases (Fig. S3D, E). The 20 factor

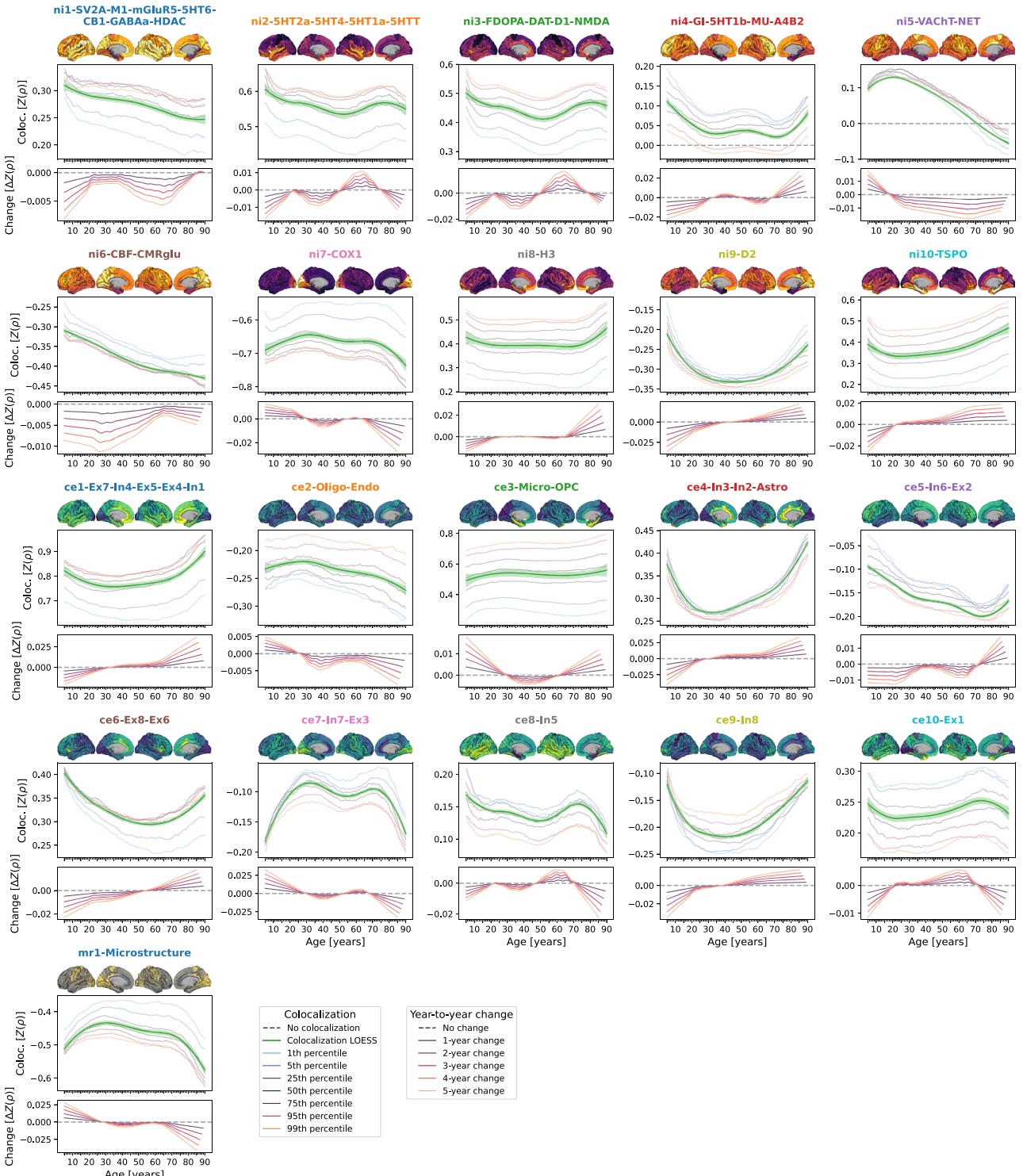

scores, in addition to the marker of cortical microstructure (mr1), represented the neurobiological markers to be evaluated in subsequent main analyses (surface plots in Fig. 2 and Fig. S4). The dimensionality-reduced markers represented biologically meaningful entities with the first factor capturing the first spatial component of cortical transmitter systems (ni1), followed by more specific factors broadly representing serotonergic (ni2), dopaminergic (ni3, ni9), and cholinergic systems (ni5) as well as brain metabolism (ni4, ni6) and immunity (ni7, ni10). Similarly, mRNA expression-derived factors entailed one general neuronal

dimension (ce1) and several more specific excitatory and inhibitory neuronal (ce4–10) and glial factors (ce2–3).

### Mapping neurobiological markers to cortical development

In the following, we report on how these neurobiological markers colocalize and explain CT change patterns between 5 and 90 years of age (Fig. 1B), spanning developmental periods from "middle and late childhood" to "late adulthood" as defined previously[1,27] (see following Figs.). CT trajectories for 148 Destrieux regions were derived from a normative model of CT development[3] estimated from cross-sectional

**Fig. 2 | Colocalization between cross-sectional modeled CT and neurobiological markers across the lifespan.** Lifespan trajectories of colocalization between neurobiological markers and modeled cross-sectional CT. For each marker, the upper panel shows a surface projection of the parcellated data; yellow-violet: nuclear imaging markers, yellow-green: gene-expression, yellow-gray: micro-structural; yellow = higher density. The center panel shows the marker's colocalization trajectory: Z-transformed Spearman correlation coefficients are shown on the y axis, age on the x axis; blue-to-orange lines indicate percentiles of modeled CT data (see legend, note that these do not show actual percentiles of colocalization strengths); the green line (LOESS = locally estimated scatterplot smoothing) was smoothed through the percentile data to highlight trajectories (shades: 95% confidence intervals). The lower panel shows year-to-year changes (y axis) derived from the LOESS line in the upper plot. See Fig. S7 for trajectories including ABCD and

IMAGEN subjects and Fig. S8 for trajectories split by sex. Coloc. colocalization, SV2A synaptic vesicle glycoprotein 2A, M1 muscarinic receptor 1, mGluR5 meta-botropic glutamate receptor 5, 5HT1a/1b/2a/4/6 serotonin receptor 1a/2a/4/6, CB cannabinoid receptor 1, GABAa γ-aminobutyric acid receptor A, HDAC histone deacetylase, 5HTT serotonin transporter, FDOPA fluorodopa, DAT dopamine transporter, D1/2 dopamine receptor 1/2, NMDA = N-methyl-D-aspartate glutamate receptor, GI glycolytic index, MU mu opioid receptor, A4B2 = α4β2 nicotinic receptor, VAChT vesicular acetylcholine transporter, NET noradrenaline transporter, CBF cerebral blood flow, CMRglu cerebral metabolic rate of glucose, COX1 cyclooxygenase 1, H3 histamine receptor 3, TSPO translocator protein, Microstr cortical microstructure, Ex excitatory neurons, In inhibitory neurons, Oligo oligo-dendrocytes, Endo endothelial cells, Micro microglia, OPC oligodendrocyte pro-genitor cells, Astro astrocytes. Source data are provided as a Source Data file.

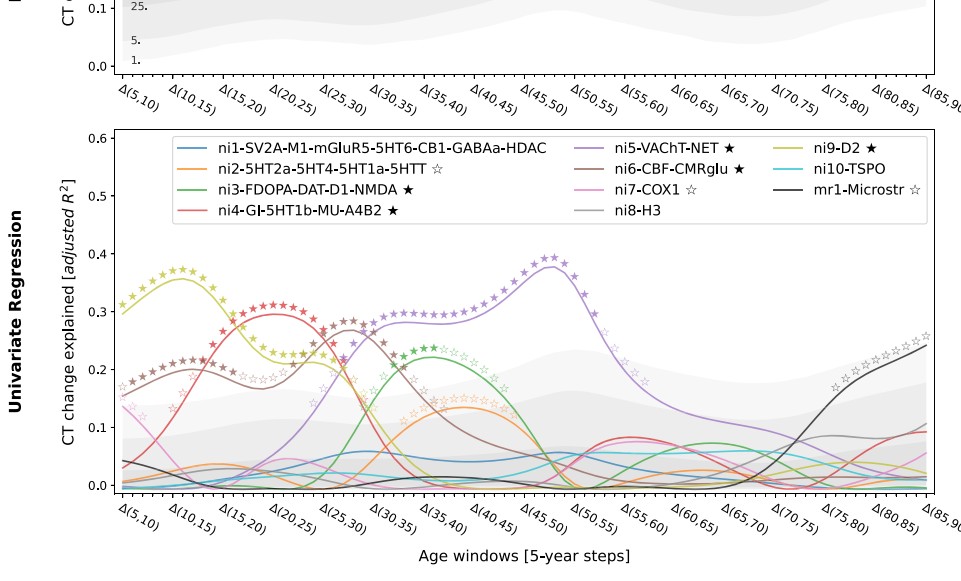

**Fig. 3 | Modeled lifespan CT change patterns explained by molecular neuro-biological markers and cortical microstructure.** Associations between modeled lifespan CT change and neurobiological markers derived from imaging modalities. Developmental periods covered by this study as defined by Kang et al.[27] are shown on top. Time periods were aligned to the center of each modeled CT change step (e.g., Δ(5,10) = 7.5). Colored lines show the amount of spatial modeled CT change variance explained (y axis) by the combined markers (upper) or each marker individually (lower) throughout the lifespan (x axis). Each y value represents the results of one multiple (upper) or single (lower) linear regression model predicting CT change across regions from neurobiological marker densities across regions.

Stars indicate positive-sided significance of each regression model based on null regression models estimated on permuted marker maps; ★: FDR-corrected across all tests shown in each panel of the plot; ☆: nominal $p < 0.05$. To provide an esti-mate of the actual observed effect size, gray areas show the distributions of modeled CT change explained by permuted marker maps ($n = 10,000$). For the lower panel, null results were combined across marker maps. See Fig. S6C for all CT change maps, and Fig. S4 for all predictor maps. CT cortical thickness, PET positron emission tomography, MRI magnetic resonance imaging, FDR false discovery rate, see Fig. 2 for abbreviations used in neurobiological marker names. Source data inlcuding exact p values are provided as a Source Data file.

data of over 58,000 subjects (from here on referred to as "modeled CT"; Fig. 1B; Text S1.3.1; age distribution: Fig. S5; CT trajectories: Fig. S6A and Movie S1). First, we tested if modeled cross-sectional CT at each given time point in life was distributed across the cortex in patterns reflecting the distributions of specific neurobiological markers

(Fig. 1C)[32]. To further understand the observed spatial associations, we then followed a hierarchical analysis framework based on regression models predicting the spatial patterns of pseudo-longitudinal "mod-eled CT change" from neurobiological markers. The outcome was quantified as the overall and marker-wise explained variance $R^2$,

**Fig. 4 | Modeled lifespan CT change patterns explained by cellular neurobiological markers.** Associations between modeled lifespan CT change and neurobiological markers derived from mRNA expression data. The figure layout and shown plot elements correspond to Fig. 4. See Fig. S6C for all CT change maps, and Fig. S4 for all predictor maps. CT cortical thickness, see Fig. 2 for abbreviations used in neurobiological marker names. Source data including exact *p* values are provided as a Source Data file.

interpretable as the percentage to which (modeled) CT change patterns can be explained from neurobiological markers[25,26]. In the first set of regression analyses, we assessed the combined and individual relevance of all 21 neurobiological markers for CT development and aging (D). In the second step, after identifying a subset of significantly associated markers, we evaluated their role in jointly explaining modeled CT changes while accounting for shared variance (E). In the final regress step, we repeated the analyses with those original neurobiological markers (i.e., before factor analysis) that loaded most strongly on the identified subset to demonstrate the validity of the factor-level markers. Next, we utilized developmental gene expression data (F) to validate and further specify our imaging-based findings (G). Last, we transferred our approach to longitudinal CT data from approximately 8,000 adolescents[47,48] (H) to demonstrate that time period-specific association patterns identified using the normative model translate to the individual subject level (I).

## Cross-sectional modeled CT shows diverse colocalization trajectories
Structural patterns resulting from the relative distribution of CT across cortical regions vary depending on the time point in life[49]. Temporal changes of these patterns might mirror the contribution of a certain cellular or molecular process to CT changes at a given time point. Using spatial Spearman correlations between each neurobiological marker and modeled CT at each time point, we revealed diverse colocalization trajectories with a general pattern of strongest changes

from childhood to young adulthood (up to approximately 30 years) as well as in late adulthood (from 60 years onwards; Fig. 2 and S7). Colocalization strengths varied across the modeled CT percentiles extracted from the normative model, but temporal trajectories were consistent. On visual comparison, trajectories appeared similar across sexes but partly differed in overall colocalization strength (Fig. S8). The modeled nature of the colocalization estimates precluded statistical tests, which would need to be conducted in individual-level data.

## Neurobiological markers explain CT change
Studying population-level and individual brain development and aging inevitably requires looking at respective changes over time, rather than focusing only on cross-sectional data[50]. We now asked to which extent different neurobiological markers explained the relative modeled change of CT across the lifespan (Figs. S6B, C) and which markers showed the strongest associations. Multivariate regression analyses predicting modeled CT change across 5-year periods throughout the lifespan (sliding window with 1-year-steps) showed that the combined, either molecular- or cellular-level, markers explained up to 54% of the spatial variance in modeled CT changes with peaks during young adulthood (molecular, 20–35 years) and adolescence (cellular, 15–20 years) [false discovery-rate (FDR)-corrected; Figs. 3 and 4, top]. Combining all 21 markers across biological levels explained up to 67% of modeled CT changes during the adolescence-to-adulthood transition (Fig. S9). Individually, 9 of the 21 neurobiological markers explained up to 15–38% of modeled CT change patterns, with most markers showing

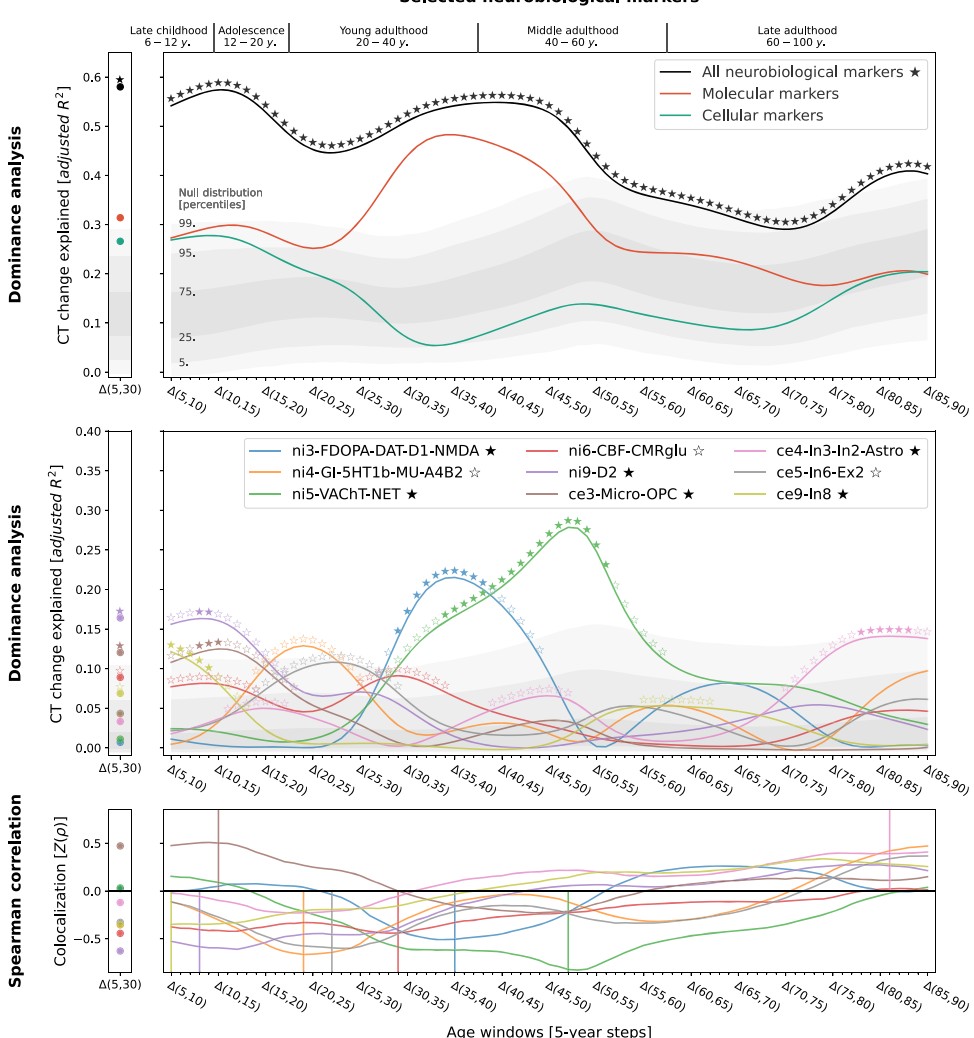

**Fig. 5 | In-depth analysis of the neurobiological markers most relevant for explaining modeled CT change patterns across the lifespan.** Modeled lifespan CT change explained by neurobiological markers, selected from the univariate analyses (Figs. 3 and 4; 9 FDR-corrected significant markers). See Fig. 3 for descriptions of global plot elements. Top: overall explained modeled CT change variance, the two colored lines highlight contributions of molecular and cellular markers. Middle: Marker-wise contributions to the overall explained spatial variance. Note that, as the used total dominance statistic describes the average $R^2$ associated with each predictor relative to the "full model" $R^2$, the sum of the predictor-wise values at each timepoint in the middle plot equals the $R^2$ values expressed in the upper panel. Bottom: Spearman correlations between modeled CT change and markers to visualize the sign of the association patterns. CT cortical thickness, see Fig. 2 for abbreviations used in neurobiological marker names. Source data including exact *p* values are provided as a Source Data file.

peaks up to young adulthood, i.e., between 5 and 30 years of age (FDR-corrected; Figs. 3 and 4, bottom). These 9 markers represented major neurotransmitters (dopaminergic, glutamatergic, cholinergic, noradrenergic), features of brain metabolism, neuron populations, and glial cells. All findings were robust against correction for baseline modeled CT as well as changes in sliding window step size, modeled sex, and modeled CT percentile (Figs. S9 and S10). Results were consistent after projecting all surface data into the more often used, but low-spatial-resolution, Desikan-Killiany parcellation[51] (Text S1.3.2, Fig. S11). We did not find evidence for confounding effects of (i) the age distribution of the lifespan sample from which the modeled CT data was obtained or (ii) the approximated average ages of the adult samples from which neurobiological markers were derived (Text S1.3.3, Figs. S12, S13).

**Specific neurobiological markers drive explained CT change**
Next, we sought to understand in detail how the 9 significant neurobiological markers contributed to the overall explained modeled CT change while accounting for correlation and shared spatial variance

patterns between molecular and cellular levels. Given that we found both the strongest modeled CT changes and CT associations during the period from childhood to young adulthood and given the particular clinical relevance of this timespan, we included modeled CT change from 5 to 30 years as an additional time window for further testing. Using dominance analyses[25,26,52] to quantify the individual contribution of each univariately FDR-significant neurobiological marker in a multivariate setting, we found that the 9 molecular and cellular markers jointly explained 58% of modeled CT change patterns from 5 to 30 years, peaking at the transition from childhood to adolescence (10–15 years; Fig. 5, top). All 9 neurobiological markers contributed to the overall explained modeled CT change during different life periods (nominal *p* < 0.05) with 6 markers surviving FDR correction (Fig. 5, middle; Movie S2). During childhood and adolescence, 3 of these 6 markers explained most of the modeled CT change patterns, representing estimates of dopaminergic receptors (ni9; $R^2 = 16\%$; peek at 8–14 years), microglia and oligodendrocyte progenitor cells (ce3; $R^2 = 12\%$; 8–15 years), as well as of somatostatin-expressing interneurons (ce8; $R^2 = 12\%$; 5–14 years). Modeled CT change patterns

during young and middle adulthood were explained by 2 neurobiological markers broadly associated with the major – i.e., dopaminergic, glutamatergic, cholinergic, and noradrenergic – neurotransmitter systems (ni3 and ni5; 29–56 years). Finally, late adulthood-modeled CT aging patterns were associated with a marker representing inhibitory neuron populations and astrocytes (ce4, 78–88 years). Except for microglia and oligodendrocyte progenitor cells, all identified associations were negative, i.e., indicating a stronger reduction of modeled CT in areas with higher density of the respective biological marker.

### Specific cortical regions drive CT change associations

The spatial associations reported here are likely dominated by some cortical regions relative to others. By evaluating the impact of iteratively excluding each region from the multivariate models, we found that the most influential regions differed depending on the markers. For example, cellular markers associated to childhood and adolescence modeled CT development (ce9: somatostatin-expressing interneurons and ce3: microglia) were driven by premotor, cuneus, and frontopolar areas, whereas the association to dopaminergic receptors during this period (ni9) was more influenced by primary visual, midcingulate, and insular regions. While associations between modeled CT change during young and middle adulthood and cholinergic neurotransmission (ni5) exhibited similar patterns, adult colocalization to dopaminergic neurotransmission (ni3) was strongly influenced by sensorimotor areas (Fig. 6; Text S1.3.4; Fig. S14; Movie S2).

### Factor-level markers reflect original brain atlases

Thus far, we focused on a lower-dimensional representation of neurobiological markers, which reduced predictor intercorrelation and increased statistical power, as compared to using the original 49 brain atlases. Nevertheless, we found that original atlases that were most closely related to each factor explained modeled CT change patterns to a similar extent as the factor-level models, aiding interpretation and supporting the validity of the factor-level approach (Text S1.3.5; Fig. S15 and S16). All univariate spatial associations between modeled CT change and the tested original atlases reached nominal significance ($p < 0.05$). Separate dominance analyses for each factor-level neurobiological marker with only strongly loading original atlases as predictors confirmed contributions of specific original atlases to the factor's peak explained variance: somatostatin-expressing interneurons, dopaminergic D1 and D2 receptors, as well as glucose metabolism and aerobic glycolysis accounted for most of the associated markers' peak effects during childhood and adolescence (ce9, ni9, ni4, and ni6). Peak effects during young and middle adulthood were mostly accounted for by α4β2 nicotinic receptors and the acetylcholine transporter (ni5) as well as the glutamatergic NMDA receptor (ni3; Fig. S15).

### Developmental gene expression supports CT change associations

Next, we turned to developmental gene expression[27] to confirm that the biological processes we found associated with cortical development were indeed upregulated in the cortex during the identified developmental period[53]. From a human postmortem dataset ($n = 33$, age range 0.33–82.05 years, see Kang et al.[27] for details), we estimated gene expression trajectories across the neocortex corresponding to each original brain atlas relevant for the final 9 factor-level neurobiological markers (c.f., Fig. S15). For cell-type atlases, we averaged normalized gene expression values across the respective marker genes[42,43]. For molecular markers, we selected genes corresponding to each protein(-compound), in addition to two sets of genes associated with brain metabolism[54] (Data S1). To pose as little assumptions on the sparse data as possible, we compared each gene/gene set during the age period with which the respective marker was associated with a control set of non-brain genes, testing (i) if the gene/gene set showed

higher mean expression and (ii) if it showed a peak in its trajectory, quantified as a higher ratio of expression during versus outside the age period. As expected, most genes/gene sets showed higher mean expression and/or higher expression ratios during the respective neurobiological marker's peak period, indicating that they were active in cortical tissue or had their individual peak expression in these timeframes (FDR-corrected; Fig. 7 and S17). Conversely, we observed relatively stable phases during mid-adulthood for most genes/gene sets, supporting our mechanistic rationale (see Introduction). Notably, for the two molecular markers explaining modeled CT aging patterns in adulthood (ni3: glutamatergic/ dopaminergic and ni5: cholinergic/ noradrenergic), we found evidence for associations only with dopaminergic D1 and glutamatergic NMDA receptors for ni3 as well as with the cholinergic α4β2 receptor for ni5.

### Neurobiological markers explain individual CT trajectories

The above analyses successfully demonstrated that specific neurobiological markers account for a large proportion of variance arising from modeled CT change patterns. During the neurodevelopmental period from childhood to young adulthood, 6 markers accounted for about 50% of the total variance with D1/2 dopaminergic receptors, microglia, and somatostatin-expressing interneurons taking the largest share. Relevance of all these 6 markers during their respective associated neurodevelopmental periods could be confirmed in independent gene expression data. However, a sole focus on modeled population CT change, i.e., median predictions from the normative model[3], does not allow for inferences about individual-level neurodevelopment, which is the mandatory prerequisite for exploring potential sources of interindividual variability. A successful validation in individual longitudinal data can also strengthen the potential mechanistic relevance of the identified neurobiological markers and support the use of normative models to non-invasively study neurodevelopmental mechanisms.

To demonstrate that our approach could be transferred to the individual level, we obtained 2-to-8-year longitudinal data from two large multi-site cohorts[47,48], covering the neurodevelopmental period from late childhood to young adulthood (ABCD Study®: $n = 6789$; IMAGEN: $n = 915$–1142; Demographics and quality control: Text S1.4.1, Data S2, Fig. S18). Notably, only the ABCD baseline data, but not the ABCD follow-up data or any IMAGEN data were used in estimation of the Rutherford et al. CT model[3]. To further ensure independence from the CT model, we independently harmonized the ABCD and IMAGEN CT data across sites using ComBat(-GAM)[55,56] (Tabs. S3 and S4). To nevertheless provide a reference for the extent to which CT changes could be explained in independent longitudinal data as compared to corresponding CT model predictions, we projected the ABCD and IMAGEN data into the CT model (Text S1.4.1) and extracted predicted CT for each individual subject and session (observed-vs.-predicted CT change patterns and correlations: Figs. S19 and S20). We first confirmed that the colocalization between cross-sectional single-subject CT and neurobiological markers mirrored the patterns observed for the modeled population-average (Figs. S7, S8, and S21). In line with these findings, the cohort-average relative change of CT across study timespans (appr. 10–12, 14–22, 14–19, and 19–22 years) was explained to extents comparable with predictions by the normative model (minimum/maximum observed $R^2 = 27/57\%$, model-prediction $R^2 = 47/56\%$; Fig. 8, upper and middle). These patterns translated to the individual-subject level, explaining on average between 9 and 18% in individual CT changes with considerable variability (range $R^2 = 0$–61%; Fig. 8, lower; Fig. S22). Looking at individual marker-wise contributions, we again found the model-based patterns to be reflected on both cohort-average and individual-subject levels (Fig. 9; Figs. S22 and S23). While the neurobiological markers predicted to be most important (D1/2 and microglia) indeed explained significant amounts of CT change, two other markers, which primarily reflected aerobic

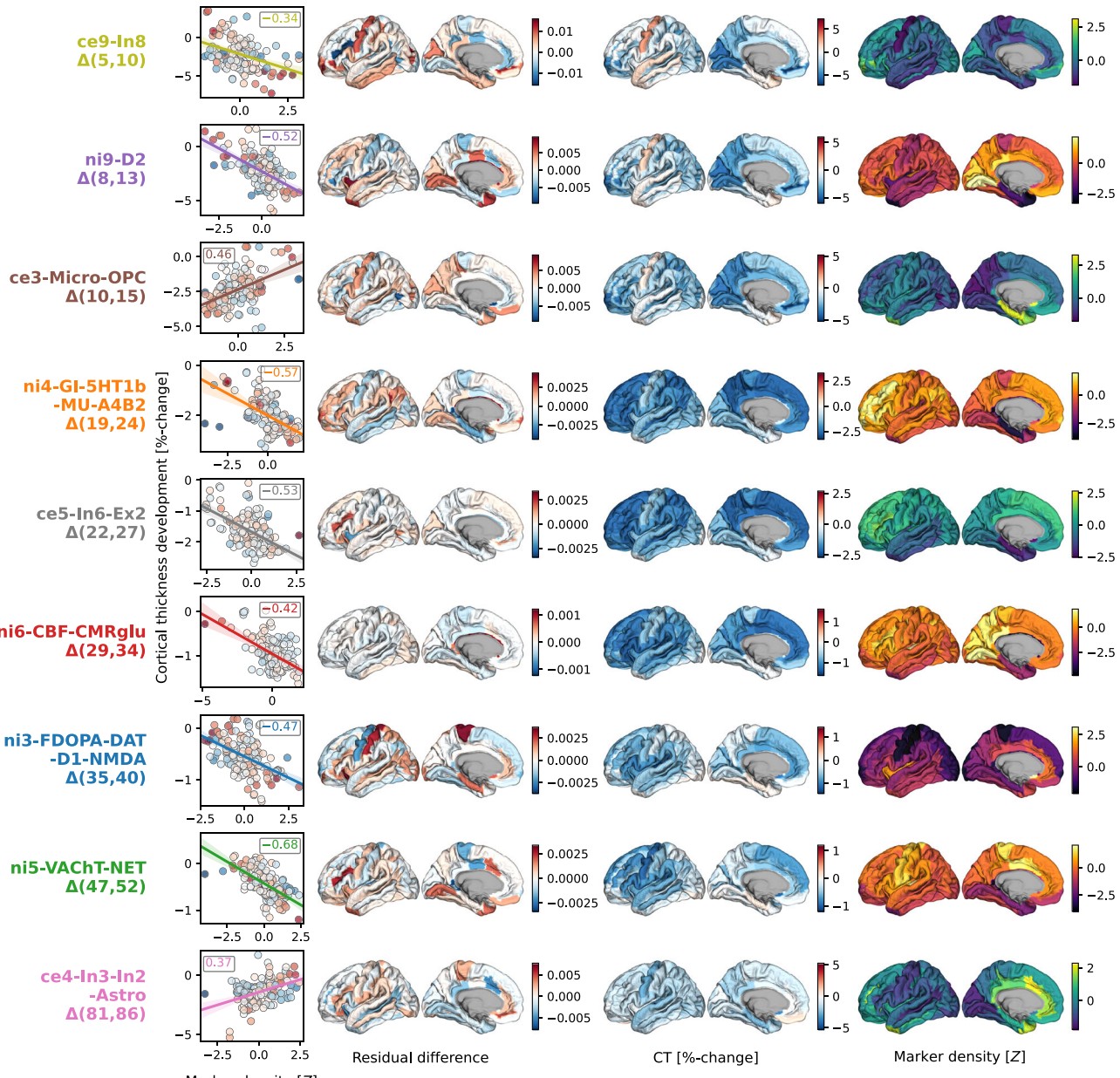

**Fig. 6 | Cortex-regional influences on modeled CT change patterns explained by most relevant neurobiological markers.** Regional influences on explained modeled CT change. Each row shows one of the 9 markers included in dominance analyses. Scatterplots: Correlation between modeled CT change at the respective predictor's peak timestep (y axis) and the predictor map, corresponding to panel A-bottom. The first surface shows the residual difference maps calculated for each marker, highlighting the most influential regions on modeled CT change association effects. For illustration purposes, the second and third surface show modeled CT change and the spatial distribution associated with the marker. Colorbars map (i) residual difference, (ii) percent-change, and (iii) z-transformed marker density; individual colorbars were not labelled to maintain readability. See Fig. S14 for all residual difference maps, Fig. S6C for all modeled CT change maps, and Fig. S4 for all predictor maps. CT cortical thickness, see Fig. 2 for abbreviations used in neurobiological marker names. Source data are provided as a Source Data file.

glycolysis (ni4) and granule neurons (ce5), were equally dominant. Sensitivity analyses showed that CT change predictions (i) generalized from the normative data to individual subjects with above-chance performance but were a poor fit for many individuals, underscoring our focus on individual differences (Text S1.4.2; Fig S24), (ii) were not relevantly influenced by ComBat or CT model-based site harmonization (Figs. S22 and S23), (iii) increased with longer follow-up duration within each time period (Fig. S25), (iv) varied by sex and study site in some tested time periods despite site harmonization of the original cross-sectional CT data (Text S1.4.3; Fig. S26), and (v) varied with

individual atypical CT development as well as data quality (Text S1.4.4; Fig. S27).

## Discussion

Patterns of spatial colocalization between macroscale brain structure and the underlying neurobiology provide in vivo insight into healthy and pathological processes that are otherwise inaccessible to human studies. Our results suggest that the spatial alignment between modeled lifespan changes of CT and corresponding adult-derived neurotransmitter receptor, brain metabolism, and cell type profiles closely

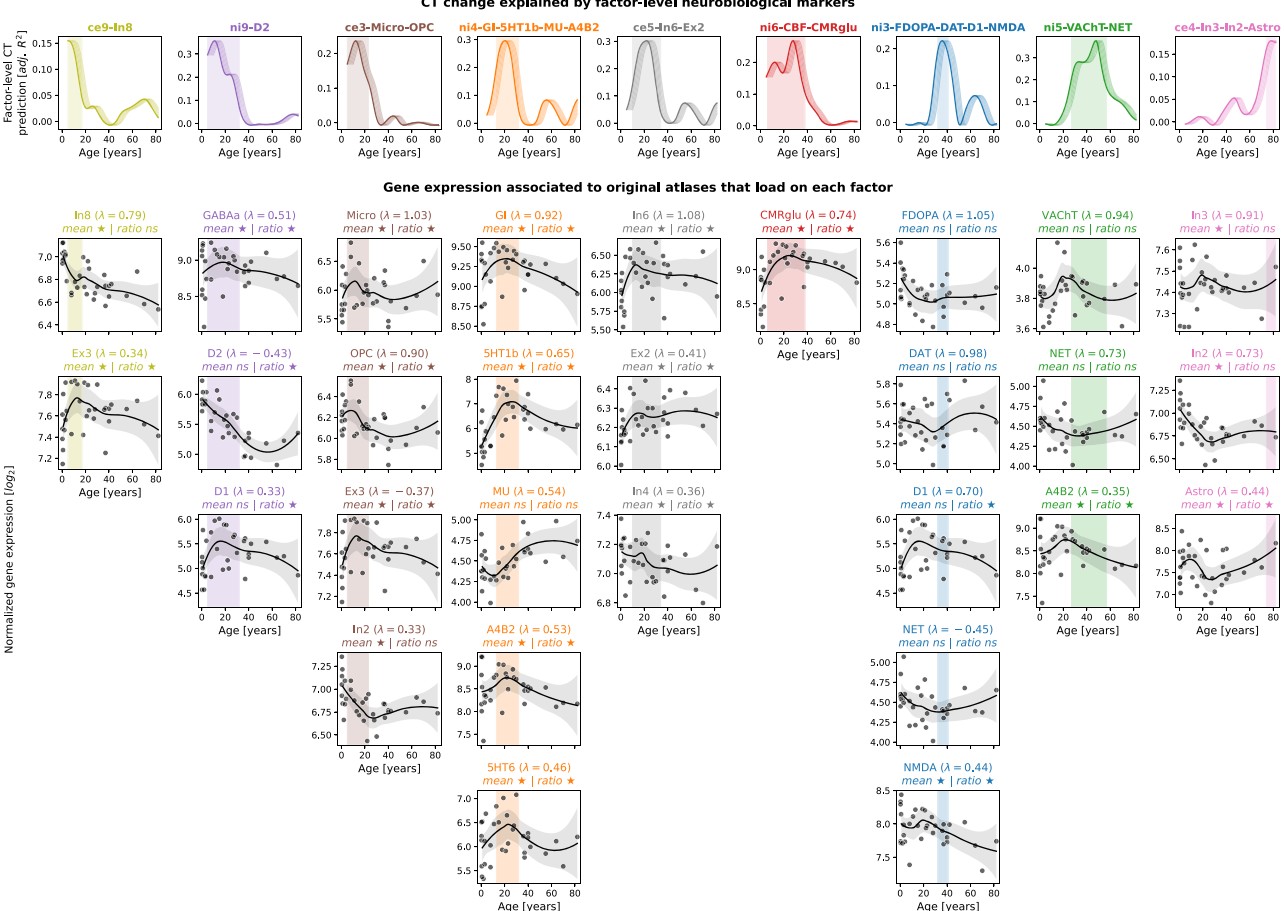

**Fig. 7 | Validation of CT model-based results in developmental gene expression data.** First row: Modeled CT change explained by individual neurobiological markers, exactly corresponding to univariate results in Figs. 3 and 4. X values are aligned to the first year of each tested modeled CT change time period (e.g., Δ(5,10) is aligned to 5 years on x-axis). Shades following each line visualize other possible alignments (Δ(5,10) is aligned to 6, 7, 8, 9, or 10 years). Vertical shaded boxes indicate time periods in which CT change was explained significantly (FDR). Following rows: Normalized log$_2$-transformed gene expression trajectories for maximally 5 original atlases that loaded on factor-level neurobiological markers with λ > |0.3| (c.f., Fig. S15). Gene expression for each marker was derived from related single genes or from averaging across gene sets. Grey dots indicate the average neocortical expression of individual subjects. Black lines and shades show locally estimated scatterplot smoothing (LOESS) curves with 95% confidence intervals. Associations were tested for by averaging the LOESS data within and outside of each respective time period and comparing mean and ratio against null data randomly sampled from non-brain genes (positive-sided exact p values). ★: FDR-corrected across all tests; ☆: nominal p < 0.05. CT CT change, adj. adjusted, FDR false discovery rate, ns not significant, see Fig. 2 for abbreviations used in neurobiological marker names. Source data including exact p values are provided as a Source Data file.

reflects neurodevelopmental processes across various neurobiological levels (see Fig. 10, S28, and Data S5 for a descriptive overview). While synaptogenesis and neuronal and glial proliferation continue into the first postnatal years, the second and third life decades are marked by a targeted reduction of neurons and cell components, likely reflecting functional specialization[10–13,27,30,57–59]. Indeed, our findings reveal dynamic patterns of associations between early CT development and neurobiological markers, in line with a diverse prior literature. Microglia have been implicated in synaptic remodeling[7,32,35] and in myelination[60,61], which has been shown to continue into young adulthood[31,34,62–64]. Brain metabolic demand was shown to peak close to the childhood-to-adolescence transition[65–67], potentially connected to puberty-associated hormonal changes[68]. Somatostatin interneuron markers were shown to remarkably decrease within the first decade of life[59]. Finally, dopamine D1 receptor activity was reported to peak in adolescence and young adulthood before declining steadily with age[69–71]. Notably, we identified the dopamine D1 and D2 receptors as the only neurotransmitter distributions that explained early CT development, in line with their initially discussed regulatory effects on cortical development[17–19]. Approaching adulthood, cortical development becomes less dynamic with most regions taking stable or steadily

decreasing aging trajectories[1,3]. Only the cholinergic system consistently predicts CT changes throughout adulthood, potentially pointing to its role in healthy and pathological aging[72]. From a broader perspective, regional patterns of cortical development were often described in terms of a segregation in uni- and transmodal brain regions, showing distinct developmental trajectories and (micro-) structural profiles[36,73]. While we indeed observe regions typically implicated in these contexts to strongly contribute to child-to-adulthood developmental colocalizations (i.e., motor and medial occipital vs. lateral prefrontal and parietal cortices), we cannot report on a clear pattern, requiring further follow-up study.

While converging findings between our spatial association analyses and prior multi-disciplinary research can be taken as confirmation for our results, these convergences can neither prove that our results actually reflect cortex-developmental neurobiology, nor can they provide explanations for non-matching results. For instance, while one might intuitively bridge from our findings on a potential connection between CT aging patterns and cholinergic, glutamatergic, and dopaminergic neurotransmitter systems to neurodegeneration, it is much harder to find confirmation for a potential mechanism in the literature, considering that our findings are based on (modeled) typical

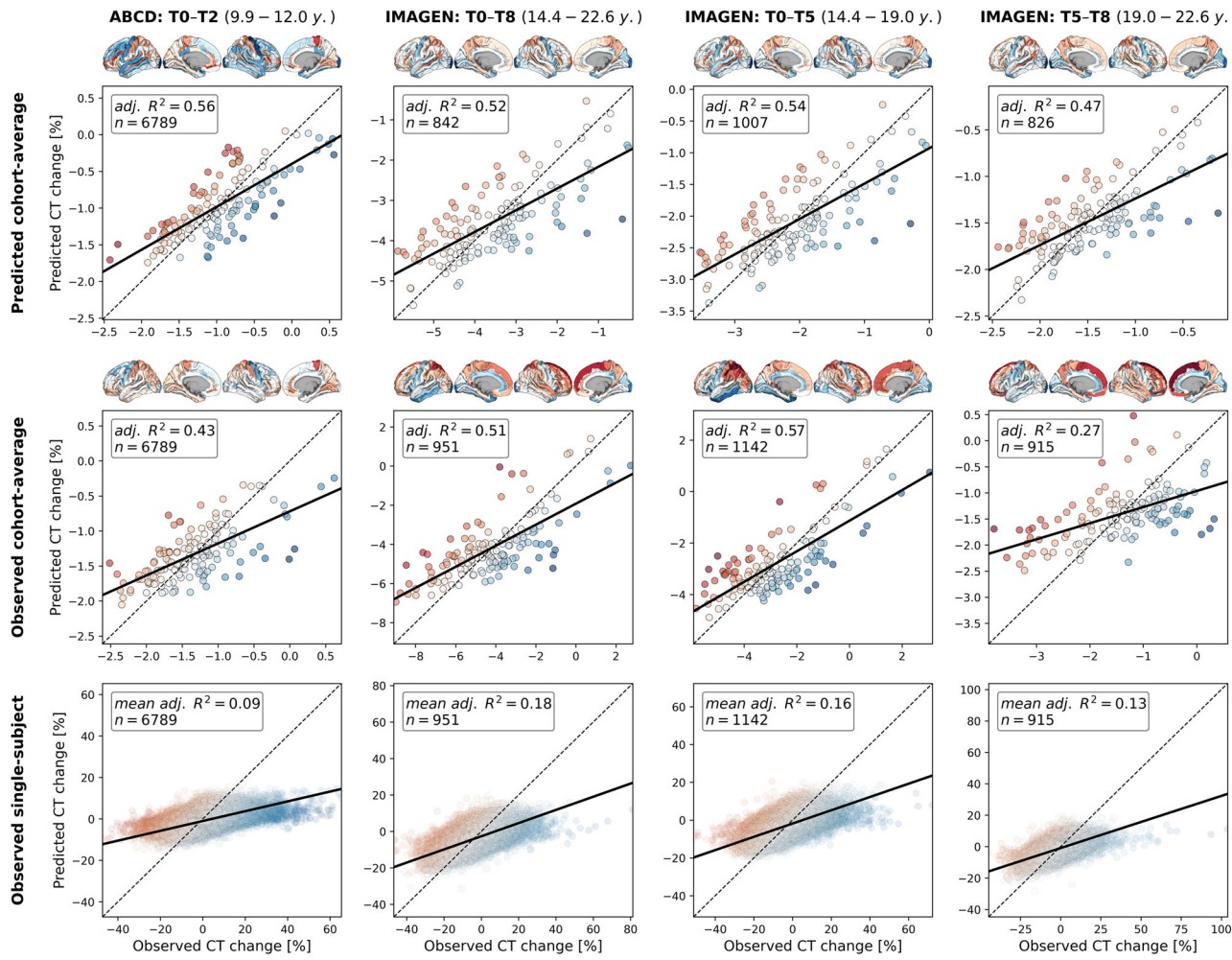

**Fig. 8 | Validation of overall explain CT change in ABCD and IMAGEN datasets.** Explained spatial CT change variance in ABCD and IMAGEN data. The overall model performance is illustrated as scatter plots contrasting predicted CT change (y axis) with observed CT change (x axis). Scatters: single brain regions, color-coded by prediction error. Continuous line: linear regression fit through the observations. Dashed line: theoretical optimal fit. Brains: prediction errors corresponding to scatters. Rows: upper: cohort-average predicted by the reference (Braincharts) model, lower sample size due to subjects dropped during model adaptation (see Methods); middle: observed cohort-average (ComBat-harmonized); lower: observed single-subject values (ComBat-harmonized), one regression model was calculated for each subject, but the results were combined for illustration purposes. CT cortical thickness, adj. adjusted. Group-level source data are provided as a Source Data file.

CT aging patterns. Interpretation of our results is furthermore complicated by the constraints of the underlying data, with, on one side, the neurobiological brain atlases, which (i) were derived from independent adult populations of varying age and sex, (ii) were processed with different strategies, (iii) were in part – as was the developmental gene expression data – obtained from postmortem samples[39], and (iv) exhibited shared spatial patterns, limiting specificity of single spatial colocalization estimates. On the other side, the normative CT model was based on (i) cross-sectional data from a (ii) largely White, Central European or North American[74] population[3,75] and (iii), while it was estimated in a sufficiently large[20] and sex-matched cohort, may yet be biased by a non-uniform age distribution. Generally, although colocalization patterns were similar when evaluated using a coarser anatomical cortex parcellation, we recommend future studies to explore finer divisions based on cortical cytoarchitecture, which could potentially reveal more detailed associations. Nonetheless, we consider the high replicability of the observed associations, despite the noise introduced by the above limitations, to rather strengthen the robustness of our findings.

The spatial colocalization framework constitutes a powerful and flexible tool to study the biological underpinnings of both typical and atypical human brain processes in vivo. While its application in a wide range of neuroimaging contexts has brought valuable insights into the neurobiology of structural[31,32,76] and functional[77] brain development, as well as brain organization in health[25] and disease[26,40,41], inferences from spatial colocalization analyses are usually limited to an associative, non-causal level. Notably, this also is the case for our present study, in which all reported associations can only provide indirect evidence for involvement of specific neurobiological markers in CT development and aging, not providing actual mechanistic explanations. We identify two pathways to further evolve the spatial colocalization approach. First, spatial colocalizations, as readily available and efficiently applicable tools, can serve as potential guidance to targeted (causal) follow-up studies of specific processes. However, experimentally establishing causality (as in: manipulation of $X$ causes changes in cortical morphology) is again likely hard to realize in humans, especially when brain morphology as compared to function is the outcome of interest. Second, from a neuroimaging perspective, the validity, meaningfulness, and – potentially – causality of spatial colocalizations can be tackled (i) by specifically testing on the individual level if the spatial distribution of a

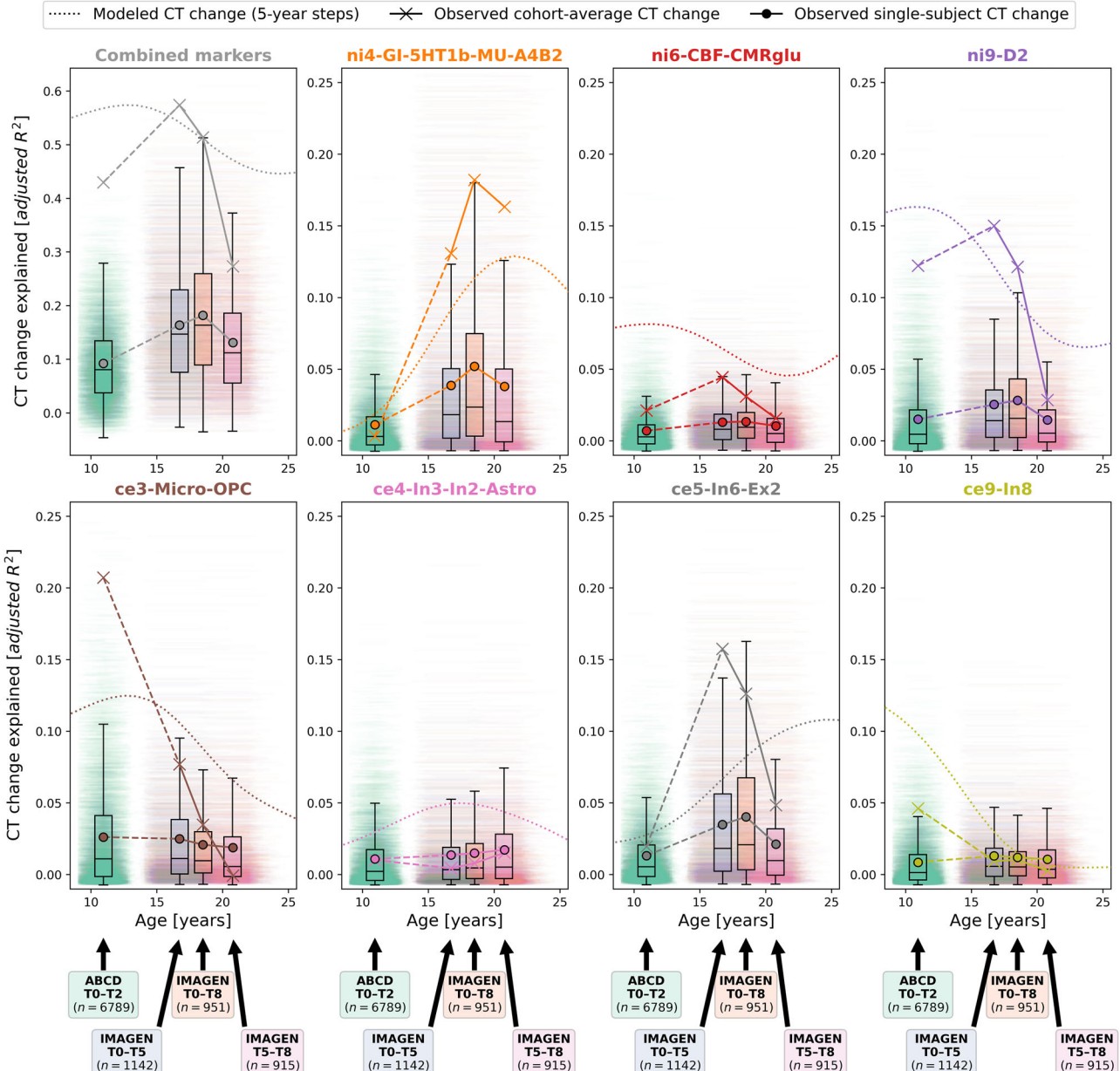

**Fig. 9 | Validation of individual marker-level results in ABCD and IMAGEN datasets.** Explained spatial CT change variance with a focus on the individual neurobiological markers. Subplots for the combined analysis and each individual marker show: modeled CT change as presented in Figs. 3 and 4 (dotted line); observed cohort-average CT change (cross markers); and observed single-subject CT change (boxplots and dot markers). For each subject, one horizontal line at their individual $R^2$ value ranges from their age at the beginning and end of each time span. Boxplots show the distribution of individual values for each time span (boxes: lower-bound: 25th, center: 50th, upper-bound: 75th percentile; whiskers: 1.5 × interquartile range). Note that the first subplot (Combined markers) corresponds to the data presented in Fig. 8. See Figs. S22 and S23 for detailed results. CT cortical thickness, see Fig. 2 for abbreviations used in neurobiological marker names. Group-level source data are provided as a Source Data file.

neurobiological process that is ethically measurable in vivo can predict the spatial distribution of neuroimaging outcome of interest, (ii) by providing supporting evidence from other levels of biological organization, which we assume to either influence or be influenced by spatially organized brain-processes, and (iii) by harnessing disorders with known pathobiology in a quasi-experimental lesion-mapping-like setting to explore influences of the pathological process on brain organization. Applying these strategies to the present work, follow-up studies could, for example, (i) test if cholinergic receptor distributions measured with PET at one adult timepoint predict CT change patterns in later adulthood, (ii) test if individual spatial colocalizations scale with genetic or epigenetic markers (as influencing factors), peripheral physiology or cognition

(as influenced factors), or (iii) test if neurodevelopmental or -degenerative disorders such as attention-deficit hyperactivity disorder, psychosis, Parkinson's, or Alzheimer's diseases lead to deviations in the expected developmental colocalization patterns (i.e., here, ni9, ni3, and ni5). Finally, strong quasi-experimental human evidence for the biological validity of spatial colocalizations between cortical morphology and neurobiological processes would be provided by the study of human subjects involuntarily exposed to development-affecting drugs (e.g., cocaine[18]) or rare disorders with specific targets such as genetic deletion/duplication syndromes or (pediatric) autoimmune encephalitides. Especially the latter could, due to their rapid onset, clinical course, and known targets (e.g., NMDA, $GABA_A$, or GAD), lead to specific alteration

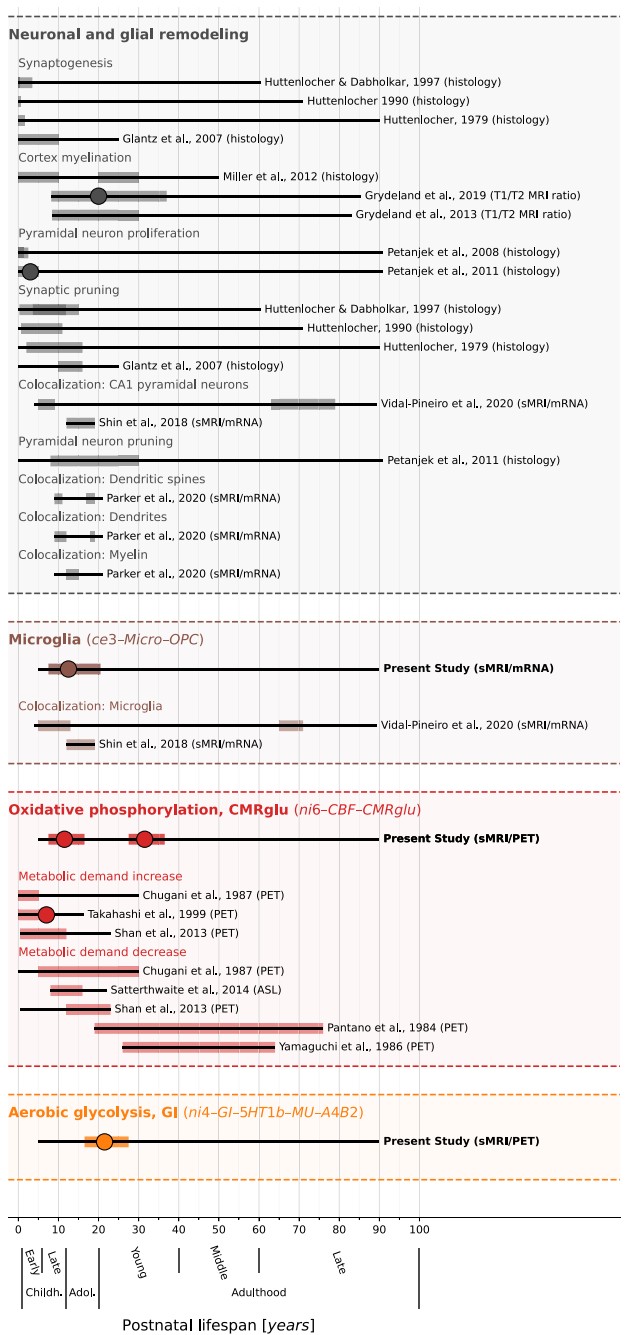

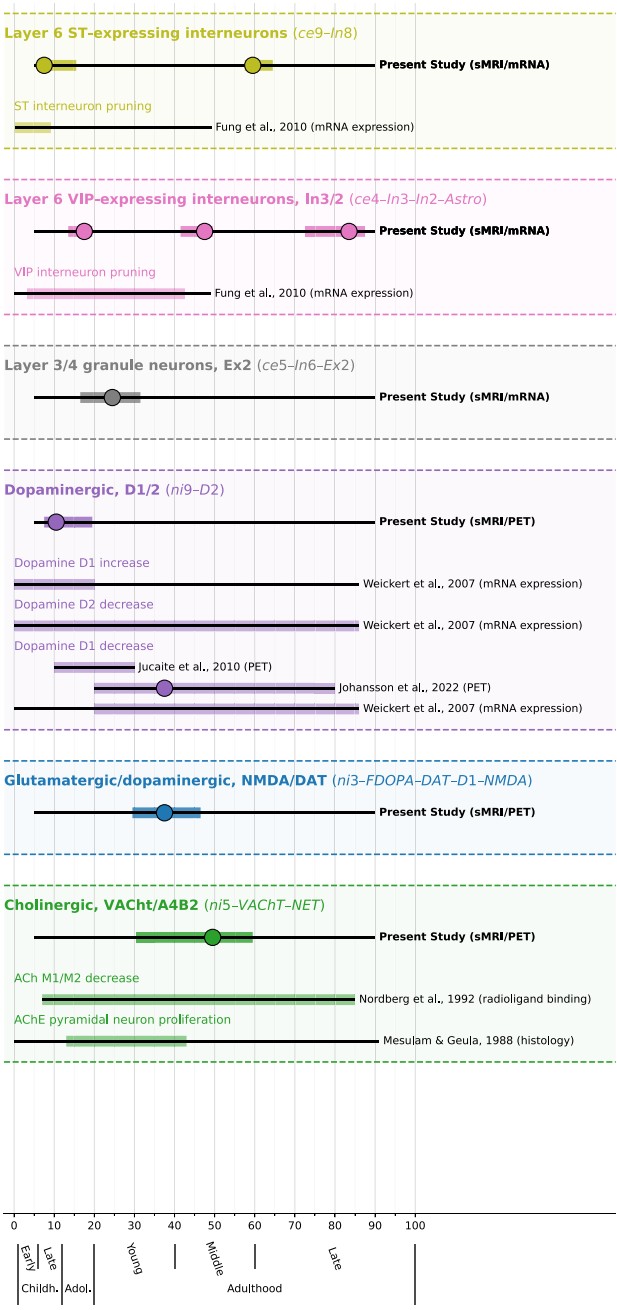

**Fig. 10 | Summary of study findings in the context of prior literature on humans.** Condensed visualization of the reported results (first line of each block, emphasized are neurobiological markers that showed consistent results) in context with related results of previous human studies investigating similar biological processes or cell populations (lines below)[10–13,30–32,35,58,59,62–69,71,109–112]. We do not claim this collection to be exhaustive. In the left upper panel, we show studies investigating general cellular remodeling processes; in the other panels, each header indicates one neurobiological marker with associated studies below. Each thin black line overlaid by a colored bar indicates results from one study. If a study reported multiple results pertaining to the same process (e.g., from two different brain regions), bars were laid over each other (Data S5 for individual listings). Thin black lines: overall time span investigated. Colored overlay: time period in which the respective study target was reported to show developmental changes (present study: nominal $p < 0.05$), independent of the sign of the association. Large dots: Timepoint of the maximum association. See also Fig. S28 and Data S5 for a more comprehensive overview including various topics. Abbreviations: ST = somatostatin, CR calretinin, sMRI structural MRI, CBF cerebral blood flow, PET positron emission tomography, ASL arterial spin labeling, ACh(E) acetylcholine (esterase), see Fig. 2 for abbreviations used in neurobiological marker names. Source data are provided as a Source Data file.

patterns of expected developmental trajectories that might be captured with spatial colocalization approaches[78,79].

Normative modeling of large-scale neuroimaging data has received considerable attention as a tool to translate basic research into clinical applications[1,3,20,22,80]. Our results indicate that if used as a reference for typical brain development, combining normative models of brain regional features with spatial colocalization approaches could facilitate discovery of physiological mechanisms underlying specific developmental patterns. Notably, here we focused exclusively on CT. Considering the diverging genetic influences and developmental trajectories of other brain-morphological features[81,82], those might show different neurobiological associations. As evidenced by our

comparative analyses between CT change predictions by the normative models and cohort-average CT changes as observed especially in the independent IMAGEN sample, normative models are capable of predicting population-level development, even when estimated only on cross-sectional data. Going beyond this group-level discovery approach, we demonstrate the feasibility of developmental spatial colocalization analyses in single subjects by mapping individual-level brain development to specific neurobiological markers. However, the strong variation in colocalization estimates observed on the individual level warrants further research into normative modeling of longitudinal data on the one side[83] and potential sources of interindividual variability on the other. Taking a clinical perspective, in view of the ability of neurobiological markers to explain typical CT development patterns, studying how these findings translate to atypical neurodevelopment[84] is a promising path for future research. Establishing developmental spatial colocalizations as potential diagnostic, prognostic, or therapeutic biomarkers will require demonstrating both their biological validity (see above) and sensitivity to the clinical outcome of interest. In the present case, their value as biomarkers may be limited by the large interindividual variation, requiring (i) further fine-tuning of individual-level cortex developmental change mapping including – if necessary – correction of study site confounds and (ii) optimization of the applied neurobiological markers in terms of source cohorts, potential dimensionality reduction, and spatial resolution.

In closing, it is important to reiterate that the central assumption of the spatial colocalization approach used here is that the spatial topology of adult neurobiology is reflected in neurodevelopmental changes in MRI-based CT estimates. Given the lack of non-invasive tools to study developmental neurobiology in vivo, our study provides evidence, albeit indirect, for such associations.

## Methods
### Ethics
No new human data were acquired for this study. Ethical approval for usage of publicly available and restricted-access databanks including human demographic, behavioral, and neuroimaging data has been granted by the Heinrich-Heine-University, Düsseldorf, Germany. Specific approval for collection and sharing of the used data (brain atlases, Braincharts model, Human Brain Transcriptome, ABCD, and IMAGEN) were provided by local ethics committees; detailed information is available in the cited sources. Informed consent was obtained from each participant and their parents in the case of underage subjects. Use of the ABCD data is registered at the NDA database at https://doi.org/10.15154/1528657. The responsible IMAGEN investigator is T. Banaschewski.

### Software
Spatial colocalizations between CT (changes) and cortical atlases were conducted using JuSpyce 0.0.2[85] (https://github.com/LeonDLotter/JuSpyce) in a Python 3.9.11 environment. Other used software[23,24,85–94] is listed in detail in Supplementary Text S1.5.

### Data sources and processing
**Atlases of molecular and cellular neurobiological markers.** Neurobiological atlases (Fig. S1) were separated into two broad categories according to their source modality. Sample characteristics and data sources are provided in Data S1.

The neuroimaging ("ni-") dataset consisted of group-average nuclear imaging atlases (neurotransmitter receptors, brain metabolism and immunity, synaptic density, and transcriptomic activity) and an MRI-based marker of cortical microstructure (T1w/T2w ratio; Text S1.2.1)[23–25,44,45,95–97]. Maps were (i) transformed from fsLR (metabolism and T1w/T2w) or Montreal Neurological Institute space (all others) to fsaverage5 space using registration fusion[24,98], (ii)

parcellated in 74 cortical regions per hemisphere (Destrieux[46]), and (iii) Z-standardized across parcels within each atlas.

Cell type ("ce-") atlases were built by (i) retrieving genetic cell type markers identified by Lake et al.[42] and Darmanis et al.[43] via single-nucleus RNA sequencing in human brain tissue from the PsychENCODE dataset[99], (ii) extracting Allen Human Brain Atlas mRNA expression values[38] for each Destrieux parcel and each marker gene using abagen[86] (default settings, data mirrored across hemispheres, Text S1.2.2), (iii) Z-standardizing the data across parcels within each gene, and (iv) taking the uniform average of the data across genes within each cell type.

We reduced the dimensionality of the atlas datasets to decrease multicollinearity in multivariate regression analyses. As the nuclear imaging and mRNA expression data likely differed strongly in terms of confounds and signal-to-noise ratio, and to study molecular- and cellular-level effects separately, data sources were not mixed during dimensionality reduction. To retain interpretability, we used factor analysis for dimensionality reduction (minimum residuals, promax rotation). All unrotated factors that explained ≥ 1% of the variance of each dataset were retained. We chose the oblique rotation method as the resulting factor intercorrelation would be expected from non-independent biological processes or cell populations. Resulting predictors were named by assigning each original atlas to the factor it loaded on the most (nuclear imaging: ni1–n; mRNA expression: ce1–n; MRI: only microstructural marker, no dimensionality reduction: mr1). In an additional analysis, we ensured that the factor solution estimated on the original brain atlases explained more variance in the original dataset than factor analyses estimated on permuted brain maps (see Text S1.2.4).

**Braincharts CT model.** The Braincharts reference model was estimated on 58,836 subjects from 82 sites (50% training/testing split; 51% female based on self-reported sex; age range 2.1–100 years; age distribution: Fig. S5). Detailed information on the included samples, CT estimation, and modeling procedure was provided by Rutherford et al.[3]. Notably, while ABCD baseline data were included in the model estimation, ABCD follow-up and IMAGEN data were not. Briefly, T1-weighted MRI data were obtained from the original cohorts and FreeSurfer 6.0[100] was used to extract parcel-wise CT data. Image quality was ensured based on FreeSurfer's Euler characteristic[101] and manual quality control of 24,354 images[3,47]. CT development was modeled separately for each Destrieux parcel using warped Bayesian linear regression models predicting CT from age, sex, and site as a fixed effect. The applied methodology was developed for use in large datasets, can model nonlinear and non-Gaussian effects, accurately accounts for confounds in multisite datasets, and allows for estimation of site batch effects in previously unseen data[3,94,102–104].

We extracted Braincharts "modeled" CT data separately for females and males for each of 148 cortical parcels for 171 timepoints (5–90 years with 0.5-year steps) and 7 percentiles (1st, 5th, 25th, 50th, 75th, 95th, and 99th). We focused on the age range of 5 years onwards as the used FreeSurfer pipeline was not adjusted for very young ages[3]. For colocalization analyses, the extracted modeled CT data were used as is. For model-based (pseudo-)longitudinal analyses, we calculated the relative modeled CT change ΔCT from year $i$ to year $j$ based on the median (50th percentile) sex-average modeled CT data as $\Delta CT_{(i,j)} = \frac{CT_j - CT_i}{CT_i}$. Lifespan CT change was then calculated using a sliding window with 1-year steps and 5-year length from 5 to 90 years as $\Delta CT_{(i,j)}, i \in [5..85] j = i + 5$.

**ABCD and IMAGEN CT data.** Processed and parcellated CT data from the Adolescent Brain Cognitive Development℠ (ABCD Study®) cohort[47] was taken directly from the ABCD 4.0 release. Baseline (T0, ~10 years) and 2-year follow-up (T2) structural MRI data were processed using

FreeSurfer 7.1.1 by the ABCD study team. Details were provided by Casey et al.[47] and in the release manual (https://doi.org/10.15154/1523041). For the IMAGEN cohort[48], T1-weighted MRI data at baseline (T0, ~14 years) and at one or two follow-up scans (T5, ~19, and T8, ~22 years) were retrieved and processed with FreeSurfer's standard pipeline (7.1.1). Following Rutherford et al.[3], we relied on the total number of surface defects as provided by FreeSurfer for quality control. We excluded subjects that exceeded a threshold of Q3+IQR×1.5 calculated in each sample across timepoints[101,105] or failed the manual quality ratings provided in the ABCD dataset. One ABCD study site (MSSM) stopped data collection during baseline assessment and was excluded. For each cohort (ABCD: $n = 20$; IMAGEN: $n = 8$ sites), we applied site harmonization to cross-sectional CT data of all subjects in one step across sessions using ComBat[56], modeling age as a non-linear covariate (ComBAT-GAM[55]) in addition to covariate effects of sex and session. Facilitating comparison between observed and Braincharts-predicted CT data, we additionally projected the ABCD and IMAGEN data into the Braincharts model to derive predictions and individual deviation scores (for sensitivity analyses, Text S1.4.1).

Colocalization analyses were calculated on the site-adjusted and original CT values at each timepoint. For longitudinal analyses, the relative CT change between each time point within each cohort was calculated as above (ABCD: T0–T2; IMAGEN: T0–T8, T0–T5, and T5–T8).

### Null map-based significance testing

Spatial associations between brain maps can be assessed in correlative analyses in the sense of testing for cortex- or brain-wide alignment of the distributions of two variables *A* (e.g., CT) and *B* (e.g., a neurotransmitter receptor)[23,25,31,106]. Effect sizes (e.g., correlation coefficients) resulting from correlating *A* and *B* convey interpretable meaning. However, parametric *p* values do not, as they are influenced by the rather arbitrary number of observations (between thousands of voxels/vertices and a few parcels) and spatial autocorrelation in the brain data[107]. Null model-based inference approaches circumvent this problem by comparing the observed effect size to a null distribution of effect sizes obtained by correlating the original brain map *A* with a set of permuted brain maps generated from *B* to derive empirical *p* values. From several approaches proposed to preserve or reintroduce spatial autocorrelation patterns in null maps[107], we relied on the variogram-based method by Burt et al.[87] as implemented in JuSpyce via BrainSMASH[24,85,87].

### Discovery analyses based on the Braincharts model

**Lifespan colocalization trajectories.** To characterize lifespan trajectories of colocalization between cross-sectional modeled CT and neurobiological markers, we calculated Spearman correlations between each brain atlas and modeled CT data at each extracted time point and percentile. Smoothed regression lines (locally estimated scatterplot smoothing) were estimated on data from all percentiles combined to highlight developmental trajectories. As the resulting developmental patterns were largely similar across sexes, we performed the main analyses on female-male-averaged modeled CT data and reported sex-wise results in the supplementary materials.

**Prediction of modeled CT change.** The main objective of this study was to determine the extent to, and the temporal patterns in which, neurobiological marker could explain modeled CT development and aging patterns. To achieve this goal, we designed a framework in which we predicted stepwise relative (modeled) CT change from one or more brain atlases in multivariate or univariate regression analyses. The amount of (modeled) CT variance explained $R^2$ was used as the main outcome measure (adjusted in multivariate analyses). Exact one-sided *p* values were calculated by generating a constant set of 10,000 null maps for each brain atlas and comparing observed $R^2$ values to $R^2$ null

distributions obtained from 10,000 regression analyses using the null maps as predictors.

To determine the general extent to which modeled CT changes could be explained, we performed one multilinear regression per lifespan timestep (81 models) using (i) all neuroimaging and (ii) all mRNA expression-based atlases. In an additional analysis, we assessed the result combining all atlases irrespective of modality. The resulting *p* values were FDR-corrected across all models and atlas source modalities. To quantify individual atlas-wise effects and identify specific neurobiological markers of potential relevance to CT development, we performed univariate regression analyses per timestep and atlas (21 × 81 models), correcting for multiple comparisons using FDR correction within each modality. In sensitivity analyses, we assessed effects of correcting for baseline modeled CT (regression of modeled cross-sectional CT at year x from CT change between year x and year y), adjusting modeled CT percentile (1st and 99th), sex (female and male separately), and window length (1-year, 2-year). As above, the results were consistent across sexes, thus all main analyses were reported for sex-average modeled CT data and the following model-based analyses were performed only on sex-average data. As our analyses are based on the 148-parcel Destrieux parcellation[46], while the low-resolution Desikan-Killiany parcellation[51] (68 parcels) is more prevalent in the literature, we evaluated if spatial association patterns remained stable using Desikan-Killiany-transformed data (Text S1.3.2). Finally, we tested if explained modeled CT change across time windows was correlated to (i) the number of subjects that went into model estimation per time window and (ii) the distance in years to the approximate age of the neurobiological marker sources (Text S1.3.3).

**Marker-wise contributions to explained modeled CT change.** Aiming to identify when and how neurobiological markers contributed to explaining modeled CT change, we retained only those brain atlases that significantly explained modeled CT development individually (FDR correction) and conducted dominance analyses predicting modeled CT change from this joint set of atlases. Dominance analysis aims to quantify the relative importance of each predictor in a multivariate regression. The total dominance statistic is calculated as the average contribution of a predictor *x* to the total $R^2$ across all possible subset models of multivariate regression and can here be interpreted as the extent to which modeled CT development during a certain time period is explained by *x* in presence of the whole set of predictors *X* and as a fraction of the extent to which modeled CT development is explained by set $X$[25,26,52]. Following from this, in our models, the sum of the atlas-level $R^2$ at a given timespan equals the total $R^2$ at this time point. The significance of dominance analyses was determined as described above by generating null distributions and estimating empirical *p* values for both, the "full model" multivariate $R^2$ and the predictor-wise total dominance $R^2$. Finally, Spearman correlations between modeled CT change and neurobiological markers were conducted to indicate the directionality of associations.

Dominance analyses were conducted at each timestep and, to highlight the main postnatal developmental period between child and adulthood, on the modeled CT development across this entire period defined as $\Delta CT_{(5,30)}$ (82 models). Resulting *p* values were corrected across the whole analysis (full model and atlas-wise: 82 + 82 × 9 *p* values).

**Brain-regional influences on modeled CT change association patterns.** To estimate the importance of individual brain regions for the associations between CT change and brain atlases, we relied on the atlas-wise residual differences across brain-regions as unitless measures of the influence of individual cortex regions on the dominance analysis results. The residual difference of prediction errors $\Delta PE$ for

each predictor $x$ out of the predictor set $X$ was calculated as $\Delta PE = \left| PE_{X\setminus\{x\}} \right| - \left| PE_X \right|$. The results were visualized on surface maps for descriptive interpretation.

**Relationships between dimensionality-reduced and original neurobiological markers.** To assess whether the factor-level markers represented the original neurobiological atlases according to the applied atlas-factor-association scheme, we performed (i) dominance analyses and (ii) univariate regressions per factor-level atlas using only the strongest associated original atlases as predictors. The latter were defined as the five atlases that loaded the most on each factor if the absolute loading exceeded 0.3. FDR correction was performed independently for dominance analyses and univariate regressions across all tests.

**Validation analyses based on developmental gene expression data**
**Data sources and (null) gene set construction.** Normalized developmental gene expression data for $n = 17,565$ genes was downloaded from the Human Brain Transcriptome database (https://hbatlas.org/pages/data); the original dataset was published by Kang et al.[27]. As of the postnatal focus of our study, we included only subjects after birth, resulting in $n = 33$, aged between 0.33 and 82.05 years. The original data was sampled across multiple cortical regions and, in some cases, both hemispheres per subject. However, because a maximum of only 11 cortex regions was sampled, we decided to average the data per subject across hemispheres and neocortical areas (c.f. Kang et al.).

We identified the original brain atlases that loaded most strongly on each factor-level neurobiological marker (c.f. section 4.5.5). Each of these original atlases was represented in the genetic data through a single gene or a set of genes (Tab. 1); in the case of gene sets, gene expression data was averaged across genes. For most nuclear imaging maps, we selected the genes or gene sets that coded for, or were associated with, the respective tracer target. For brain metabolism maps, we took two gene sets associated with aerobic and anaerobic glycolysis from Goyal et al.[54]. We did not have a gene set for the CBF (cerebral blood flow) map. For cell type maps, we took the original gene sets from which the maps were generated[42,43].

For permutation-based significance testing (see below), we created $n = 10,000$ null gene expression datasets by randomly selecting genes or same-sized gene sets from $n = 2154$ non-brain genes (https://www.proteinatlas.org/humanproteome/brain/human+brain, "Not detected in brain").

**Associations to temporal patterns of explained CT development.** The following process was used to test for associations between modeled CT change explained by neurobiological markers and developmental gene expression trajectories: (i) We fitted a smoothed regression line (locally estimated scatterplot smoothing) to the gene expression data associated with each gene/gene set as well as to the respective null gene expression datasets. (ii) For each dimensionality-reduced neurobiological marker, we identified the time period in which it explained modeled CT change significantly (FDR-corrected). (iii) For each of the (null) gene expression trajectories associated with the current neurobiological marker, we calculated the average gene expressions during and outside of the significant time period. (iv) We separately compared the mean and the ratio of gene expression during vs. outside the significant time period between the observed and null gene expression data to derive empirical one-sided $p$ values for the association between each neurobiological marker and each associated gene/gene set. (v) FDR-correction was applied across all tests at once. A significantly higher mean expression would indicate increased cortical expression of a marker during the tested time period as compared to non-brain genes. An increased ratio would broadly indicate that the marker's gene expressions peaks during the tested time period.

**Validation analyses based on ABCD and IMAGEN single-subject data**
**Developmental colocalization trajectories.** First, we tested whether colocalization patterns between neurobiological markers and single-subject cross-sectional CT followed the predictions of the Braincharts model. Spearman correlations were calculated between each subject's CT values and each atlas at all available timepoints, for CT data (i) as extracted from FreeSurfer, (ii) after ComBat-harmonization, (iii) after projection into the Braincharts model, and (iv) as predicted by the model.

**Explained CT development patterns on cohort- and individual-subject levels.** Following, we assessed how neurobiological markers that significantly explained modeled CT development during the period covered by ABCD and IMAGEN data (9–25 years) performed in individual-subject longitudinal data. Dominance analyses were performed in two steps. First, for each of the four investigated time spans (ABCD: -10–12; IMAGEN: -14–22, -14–19, 19–22 years), one dominance analysis was calculated to predict the cohort-average CT change pattern from neurobiological markers. Second, dominance analyses were calculated in the same fashion, but for every subject. For comparison, analyses were repeated on CT change patterns as predicted by the Braincharts model from each subject's age and sex. For cohort-average dominance analyses, exact $p$ values were estimated as described for the stepwise model-based analyses. For individual-level analyses, instead of estimating $p$ values for each subject, we tested whether the mean $R^2$ values of the full models and each predictor observed in each cohort and time span were significantly higher than was estimated in 1000 null-analyses with permuted atlas data. Finally, we repeated the subject-level analyses on the original CT change data prior to site-harmonization, under correction for intracranial volume, and on the longitudinal change of deviation Z scores as returned by the Braincharts model[3].

We then estimated how the subject-level regression models generalized from the subject-level normative CT change patterns to the actual observed CT change by applying the regression models estimated on each subject's normative CT change patterns to each subject's observed CT change (Text S1.4.2).

Further sensitivity analyses were conducted to estimate how CT change predictions were affected by follow-up duration, sex, study site, the prediction accuracy of CT and CT change patterns [correlation between predicted and observed CT (change), average Braincharts CT deviation (change), count of extreme deviation (change)], and data quality (number of surface defects). Subject-level full model $R^2$ values were compared by sex and study site using analyses of covariances corrected for follow-up duration (and sex or site). All other variables were correlated with full model $R^2$ values using Spearman correlations.

**Reporting summary**
Further information on research design is available in the Nature Portfolio Reporting Summary linked to this article.

## Data availability
The neurobiological marker source data, the Braincharts models, data extracted from these models, developmental gene expression data, and colocalization results generated in this study have been deposited in an associated GitHub repository (https://github.com/LeonDLotter/CTdev/; https://doi.org/10.5281/zenodo.7902901)[108]. The Braincharts models are furthermore available at https://github.com/predictive-clinical-neuroscience/braincharts. The individual-level raw and processed ABCD and IMAGEN data are available under restricted access

for data privacy reasons and by regulations of the original investigators. Access to ABCD can be obtained by application to the NIHM Data Archive (https://nda.nih.gov/study.html?id=2313). Further information on how to gain access is provided on the ABCD study website (https://wiki.abcdstudy.org/faq/faq.html). Access to IMAGEN can be obtained via a project-specific application coordinated by an IMAGEN principal investigator. The responsible researcher for the present study is author TB (tobias.banaschewski@zi-mannheim.de). Review of data access proposals is expected within weeks after submission. Protected data generated by this study can be shared with any researcher with approved data access. Group-level processed ABCD and IMAGEN data are available in the linked repository. Source Data files corresponding to each main and Supplementary Fig. are provided with this paper. As subject-level ABCD and IMAGEN data cannot be shared openly, respective Source Data files contain group-level summary metrics.

## Code availability

All code supporting the analyses and conclusions of this publication is available via the publication repository (https://github.com/LeonDLotter/CTdev/; https://doi.org/10.5281/zenodo.7901282)[108]. The code is organized in annotated Jupyter notebooks, which also produce all main and Supplementary Figs.

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

## Acknowledgements

LDL was supported by the Federal Ministry of Education and Research (BMBF) and the Max Planck Society (MPG), Germany. AS was funded by the Max Planck Society (Otto Hahn award) and Helmholtz Association's Initiative and Networking Fund under the Helmholtz International Lab grant agreement InterLabs-0015, and the Canada First Research Excellence Fund (CFREF Competition 2, 2015–2016) awarded to the Healthy Brains, Healthy Lives initiative at McGill University, through the Helmholtz International BigBrain Analytics and Learning Laboratory (HIBALL). Data used in the preparation of this article were obtained from the Adolescent Brain Cognitive Development^SM (ABCD) Study (https://abcdstudy.org), held in the NIMH Data Archive (NDA). This is a multisite, longitudinal study designed to recruit more than 10,000 children age 9-10 and follow them over 10 years into early adulthood. The ABCD Study® is supported by the National Institutes of Health and additional federal partners under award numbers U01DA041048, U01DA050989, U01DA051016, U01DA041022, U01DA051018, U01DA051037, U01DA050987, U01DA041174, U01DA041106, U01DA041117, U01DA041028, U01DA041134, U01DA050988, U01DA051039, U01DA041156, U01DA041025, U01DA041120, U01DA051038, U01DA041148, U01DA041093, U01DA041089, U24DA041123, U24DA041147. A full list of supporters is available at https://abcdstudy.org/federal-partners.html. A listing of participating sites and a complete listing of the study investigators can be found at https://abcdstudy.org/consortium_members/. ABCD consortium investigators designed and implemented the study and/or provided data but did not necessarily participate in the analysis or writing of this report. This manuscript reflects the views of the authors and may not reflect the opinions or views of the NIH or ABCD consortium investigators. The ABCD data repository grows and changes over time. The ABCD data used in this report came from https://doi.org/10.15154/1523041. This work received support from the following sources: the European Union-funded FP6 Integrated Project IMAGEN (Reinforcement-related behaviour in normal brain function and psychopathology) (LSHM-CT- 2007-037286), the Horizon 2020 funded ERC Advanced Grant "STRATIFY" (Brain network based stratification of reinforcement-related disorders) (695313), Human Brain Project (HBP SGA 2, 785907, and HBP SGA 3, 945539), the Medical Research Council Grant "c-VEDA" (Consortium on Vulnerability to Externalizing Disorders and Addictions) (MR/N000390/1), the National Institute of Health (NIH) (R01DA049238, A decentralized macro and micro gene-by-environment interaction analysis of substance use behavior and its brain biomarkers), the National Institute for Health Research (NIHR) Biomedical Research Centre at South London and Maudsley NHS Foundation Trust and King's College London, the Bundesministerium für Bildung und Forschung (BMBF grants 01GS08152; 01EV0711; Forschungsnetz AERIAL 01EE1406A, 01EE1406B; Forschungsnetz IMAC-Mind 01GL1745B), the Deutsche Forschungsgemeinschaft (DFG grants SM 80/7-2, SFB 940, TRR 265, NE

1383/14-1), the Medical Research Foundation and Medical Research Council (grants MR/R00465X/1 and MR/S020306/1), the National Institutes of Health (NIH) funded ENIGMA (grants 5U54EB020403-05 and 1R56AG058854-01), NSFC grant 82150710554 and European Union funded project "environMENTAL", grant no: 101057429. Further support was provided by grants from: the ANR (ANR-12-SAMA-0004, AAPG2019 - GeBra), the Eranet Neuron (AF12-NEUR0008-01 - WM2NA; and ANR-18-NEUR00002-01 - ADORe), the Fondation de France (00081242), the Fondation pour la Recherche Médicale (DPA20140629802), the Mission Interministérielle de Lutte-contre-les-Drogues-et-les-Conduites-Addictives (MILDECA), the Assistance-Publique-Hôpitaux-de-Paris and INSERM (interface grant), Paris Sud University IDEX 2012, the Fondation de l'Avenir (grant AP-RM-17-013), the Fédération pour la Recherche sur le Cerveau; the National Institutes of Health, Science Foundation Ireland (16/ERCD/3797), U.S.A. (Axon, Testosterone and Mental Health during Adolescence; RO1 MH085772-01A1) and by NIH Consortium grant U54 EB020403, supported by a cross-NIH alliance that funds Big Data to Knowledge Centres of Excellence.

## Author contributions

*Conception and design*: L.D.L., J.D., S.B.E. *Data acquisition and provision*: J.Y.H., B.M., C.P., T.B., F.N., G.J.B., A.L.W.B., S.D., H.F., A.G., H.G., P.G., A. Heinz, R.B., J.-L.M., M.-L. P., E.A., D.P.O., T.P., L.P., S.H., J. H.F., M.N.S., N.V., H.W., R.W., G.S., F. Nees, T., IMAGEN Consortium. *Analysis and interpretation*: L.D.L., A.S., C.P., J.D. *Manuscript drafting*: L.D.L., J.D. *Manuscript revision*: All authors. *Supervision*: J.D., S.B.E.

## Funding

## Competing interests

Dr. Banaschewski served in an advisory or consultancy role for Lundbeck, Medice, Neurim Pharmaceuticals, Oberberg GmbH, and Shire. He received conference support or speaker's fees from Lilly, Medice, Novartis, and Shire. He has been involved in clinical trials conducted by Shire and Viforpharma. He received royalties from Hogrefe, Kohlhammer, CIP Medien, and Oxford University Press. The present work is unrelated to the above grants and relationships. Dr. Barker has received honoraria from General Electric Healthcare for teaching on scanner programming courses. Dr. Poustka served in an advisory or consultancy role for Roche and Viforpharm and received speaker's fees from Shire. She received royalties from Hogrefe, Kohlhammer, and Schattauer. The present work is unrelated to the above grants and relationships. All other authors report no biomedical financial interest or other potential conflicts of interest.

## Additional information

[1]Institute of Neuroscience and Medicine, Brain & Behaviour (INM-7), Research Centre Jülich, Jülich, Germany. [2]Institute of Systems Neuroscience, Medical Faculty, Heinrich Heine University, Düsseldorf, Germany. [3]Max Planck School of Cognition; Stephanstrasse 1A, Leipzig, Germany. [4]Otto Hahn Research Group for Cognitive Neurogenetics, Max Planck Institute for Human Cognitive and Brain Sciences, Leipzig, Germany. [5]McConnell Brain Imaging Centre, Montréal Neurological Institute, McGill University, Montréal, QC, Canada. [6]Department of Neuroimaging, Institute of Psychiatry, Psychology & Neuroscience, King's College London, London, UK. [7]Discipline of Psychiatry, School of Medicine and Trinity College Institute of Neuroscience, Trinity College Dublin, Dublin, Ireland. [8]Centre for Population Neuroscience and Precision Medicine (PONS), Institute of Psychiatry, Psychology & Neuroscience, SGDP Centre, King's College London, London, UK. [9]Institute of Cognitive and Clinical Neuroscience, Central Institute of Mental Health, Medical Faculty Mannheim, Heidelberg University, Mannheim, Germany. [10]Department of Psychology, School of Social Sciences, University of Mannheim, Mannheim, Germany. [11]NeuroSpin, CEA, Université Paris-Saclay, Gif-sur-Yvette, France. [12]Departments of Psychiatry and Psychology, University of Vermont, Burlington, VT, USA. [13]Sir Peter Mansfield Imaging Centre School of Physics and Astronomy, University of Nottingham; University Park, Nottingham, UK. [14]Department of Psychiatry and Psychotherapy CCM, Charité – Universitätsmedizin Berlin, corporate member of Freie Universität Berlin, Humboldt-Universität zu Berlin, and Berlin Institute of Health, Berlin, Germany. [15]Physikalisch-Technische Bundesanstalt (PTB); Braunschweig and Berlin, Berlin, Germany. [16]Ecole Normale Supérieure Paris-Saclay, Université Paris-Saclay, Université paris Cité, INSERM U1299 "Trajectoires Développementales & Psychiatrie"; Centre Borelli, Gif-sur-Yvette, France. [17]AP-HP Sorbonne Université, Department of Child and Adolescent Psychiatry, Pitié-Salpêtrière Hospital, Paris, France. [18]Department of Psychiatry, EPS Barthélémy Durand, Etampes, France. [19]Departments of Psychiatry and Neuroscience, Faculty of Medicine and Centre Hospitalier Universitaire Sainte-Justine, University of Montreal, Montréal, QC, Canada. [20]Department of Psychiatry, McGill University, Montréal, QC, Canada. [21]Department of Child and Adolescent Psychiatry and Psychotherapy, University Medical Centre Göttingen, Göttingen, Germany. [22]Department of Child and Adolescent Psychiatry and Psychotherapy, Central Institute of Mental Health, Medical Faculty Mannheim, Heidelberg University, Mannheim, Germany. [23]Department of Psychiatry and Psychotherapy, Technische Universität Dresden, Dresden, Germany. [24]Centre for Population Neuroscience and Stratified Medicine (PONS), Department of Psychiatry and Neuroscience, Charité Universitätsmedizin Berlin, Berlin, Germany. [25]School of Psychology and Global Brain Health Institute, Trinity College Dublin, Dublin, Ireland. [26]Centre for Population Neuroscience and Precision Medicine (PONS), Institute for Science and Technology of Brain-inspired Intelligence (ISTBI), Fudan University, Shanghai, China. [27]Institute of Medical Psychology and Medical Sociology, University Medical Center Schleswig-Holstein, Kiel

University, Kiel, Germany. [28]German Center for Mental Health (DZPG), partner site Mannheim-Heidelberg-Ulm, Heidelberg, Germany.
✉e-mail: l.lotter@fz-juelich.de; leondlotter@gmail.com; juergen.dukart@gmail.com

## IMAGEN Consortium

Gareth J. Barker ⑩ [6], Arun L. W. Bokde ⑩ [7], Sylvane Desrivières ⑩ [8], Herta Flor ⑩ [9,10], Antoine Grigis[11], Hugh Garavan ⑩ [12], Penny Gowland ⑩ [13], Andreas Heinz ⑩ [14], Rüdiger Brühl ⑩ [15], Jean-Luc Martinot ⑩ [16], Marie-Laure Paillère[16,17], Eric Artiges ⑩ [16,18], Dimitri Papadopoulos Orfanos ⑩ [11], Tomáš Paus ⑩ [19,20], Luise Poustka[21], Sarah Hohmann ⑩ [22], Juliane H. Fröhner ⑩ [23], Michael N. Smolka ⑩ [23], Nilakshi Vaidya ⑩ [24], Henrik Walter ⑩ [14], Robert Whelan ⑩ [25], Gunter Schumann ⑩ [24,26], Frauke Nees ⑩ [9,22,27] & Tobias Banaschewski ⑩ [22,28]

