## [Peer Review File · Nature Communications]

Regional patterns of human cortex development correlate with underlying neurobiologyReviewer #1 (Remarks to the Author):

In this manuscript, the authors investigated spatial associations between many neurobiological signatures and lifespan changes in cortical morphology. Specifically, they collected 49 postmortem and in vivo brain atlases covering molecular and cellular processes in healthy adult samples, which were linked to lifespan changes in cortical thickness. This is a timely and important study, revealing a potential neurobiological basis for the maturation of cortical morphology. Here, I have several major and minor concerns, which are outlined below. I hope that these comments and suggestions can be used to further improve the manuscript prior to publication.

The authors hypothesized that spatiotemporal patterns of thickness development colocalize with nonpathological adult spatial distributions of the respective neurobiological markers. Given that many neurobiological signatures, such as brain metabolism, develop with increasing age, the authors should clearly address and validate the validity of spatial association analysis between lifespan changes in thickness and adult neurobiological markers.

Title: This study is a spatial association analysis and does not address how human cortex development is shaped by molecular and cellular brain systems.

Introduction. The manuscript does not provide sufficient detail on the potential mechanisms by which these molecular and cellular processes influence the development of cortical thickness. A further description would be beneficial.

The brainchart data are from a previously published study (Rutherford et al.). However, from the age distribution shown in Fig. S5, there are very different numbers of subjects at different ages. In particular, there are very few subjects before age 10 and around 40. Is the brainchart data affected by these confounding variables?

Lines 1–10, Page 6. The authors show the spatial associations between cross-sectional thickness and neurobiological features and describe diverse colocalization trajectories. However, the results are not clearly described. The authors found a general pattern of strongest changes from childhood to young adulthood (up to approximately 30 years) as well as in late adulthood (from 60 years onwards; Fig. S7). A possible reason for this result is that the neurobiological maps are from adult brains. Figure S7 should be moved to the main text. The authors stated that sex did not relevantly influence the trajectories (Fig. 8). Did they perform statistical comparisons for sex differences?

The authors showed the spatial associations between neurobiological markers and changes in cortical morphology. Several studies have demonstrated a network mechanism underlying the maturation of cortical morphology (e.g., PMID: 27457931; PMID: 38278807). I would suggest adding a discussion on this topic.

The curves presented in Fig. 2 appear to differ from those in Fig. 3A. It is unclear which result should be referenced to understand which molecular and cellular markers more significantly constrain cortical thickness development at specific ages. Please clarify.

Line 21, Page 2. Introduction. The authors describe that "neurodevelopmental disorders are associated with both deviations in brain structure and dysfunction of several neurotransmitter systems, but suffer from a lack of reproducible biomarkers and little clinical translation of neuroimaging research.". Many previous studies have reported that neurodevelopmental disorders such as autism and depression are associated with deviations not only in brain structure but also in functional connectivity. Moreover, the reproducibility of results has been well studied using multisite neuroimaging data. The authors need to rephrase their claims. In addition, this paragraph describes the importance of neurotransmitters and the normative model in understanding disease or atypical development. However, this is not reported in the results section, which is confusing.

Line 5, Page 3. Introduction. The authors described that "multimodal neuroimaging-based spatial association approaches can provide a window into specific biological mechanisms, but until now were limited to postmortem data and the cellular level. " In fact, there are many multimodal neuroimaging studies showing spatial associations between brain signatures such as brain

connectivity and metabolism (PMID: 23319644; PMID: 33443160; PMID: 30741935).

Line 30, Page 3. 21 neurobiological maps should be presented in the main text, given their importance for the association analysis. Results: Ni8 was not mentioned.

Line 18, Page 6. The authors showed that molecular- or cellular-level markers explained up to 54% of the spatial variance. How about the combined molecular and cellular-level markers? How about the sex differences?

Line 8, Page 8. In the main text, the description of Fig. 3A states, "We found that the nine FDR-significant molecular and cellular markers jointly explained 58% of CT change patterns from 5 to 30 years." However, Fig. 3A is labeled as "neurobiology markers significant in univariate analyses." This raises the question of whether the analysis is multivariate or univariate. Please clarify.

In Fig. 6, the meaning of the colored overlay is unclear. What do different shades of the same color represent (for example, in the context of synaptic pruning) (Huttenlocher & Dabholkar, 1997)?

The discussion section lacks an in-depth exploration of the physiological significance reflected by the constraint of relevant neurobiological markers on cortical maturation at specific developmental stages. This should be a central concern of the paper and needs further elaboration.

Reviewer #2 (Remarks to the Author):

The study integrates numerous "multilevel" cortical atlases, normative trajectories of MRI based cortical thickness and partly separate population-based longitudinal youth samples in an attempt to improve our understanding of the neurobiology underlying cortical thinning across the life span. I find the study comprehensive and highly relevant for a broad field within MRI based neuroimaging i.e. all studies assessing cortical macrostructure.

My comments are addressed below in chronological order and I sincerely hope they can be of some help. As this is a multidisciplinary study, I would like to state that I am a neuroscientist and refer to other reviewers to better scrutinize topics outside of neuroimaging. Also, please do not hesitate to correct me whenever I have misunderstood something!

1. Introduction: I believe reference nr. 4 regarding the link between MRI based cortical development and cognitive development from the year 2000 is dated and could be updated.
2. Introduction: I find the use of the terms "development" vs "change" somewhat unclear, particularly when attempting to grasp the underlying logic of the paper i.e: "Assuming that CT changes over the lifespan are shaped by activity, development, or degeneration of cell populations and molecular processes, it is to be hypothesized that the spatiotemporal patterns of CT development colocalize with nonpathological adult spatial distributions of the respective neurobiological marker. Increased colocalization of CT changes with an individual marker at a given developmental period would then support a role of the associated cell population or process in respective CT changes.» For the most part it appears that development is not simply "change over time", which then could pertain to the full lifespan but instead more closely relates to "maturational" processes found in youth only, is that correct? And that for the adult part of the lifespan the word "change" is often used to capture a broader set of processes? I think different disciplines use the word "development" quite differently, so maybe either define in a word or two or be very concise about usage.
3. Introduction: The example paragraph from comment 2 holds vital logic for the paper but was (to me) a heavy read, where I had to dissect every sentence. Is there a way to, when presenting this very central concept for the first time, do it "better"/simpler?
 - I think this text captures why it would make sense to map cortical thinning in youth onto adult biological architecture, but I could not fully grasp why that is not problematic. Particularly as the introduction previously states that different mechanisms might underlay youth based- and adult thinning.
 - The reader could also be guided better through this very first introduction of increased co-

localization. Am I correct in that if a marker is predominantly found in a specific region and those regions show either a more pronounced thickness change, or a change that is steady but improves in precision to the underlying distribution (the spatial patterns "match better"), or a degree of change that more closely match the "concentration" of the neurobiological marker, then all these instances would increase co-localization?

- Is the passive "it is to be hypothesized" a general statement or your actual hypothesis? Is there a suggestion of causation in this statement?

4. Introduction: I find the introduction to some of the biological terminology at times unclear. the term "(multilevel) neurobiological markers" is introduced first in regard to cellular densities and functional properties although not used too often within the text. There is also a distinction between molecular and cellular levels which is used much more frequently including within the title, and as a strength of the paper, but molecular is never formally introduced in the same way cellular is. If I understand correctly in this context "molecular brain atlases» relate mostly to neurotransmitters (and the functional level?) and are based on nuclear imaging. Is there a more structured way to present these topics? Moreover, I am wondering if "cell type"/"cell density"/"cell population", and "process"/"function" is also used interchangeably at times, but I might be wrong here, so great if the authors could have a look, but they know best.

5. Introduction: What are the practical obstacles meant here? "Our current knowledge on biological factors that guide typical human cortex development is severely limited by practical obstacles»

6. Introduction: The motivation and goal of the study appears somewhat generic and scattered within parts of the introduction. There are references to biomarkers and some general statements of psychopathology "... suffer from a lack of reproducible biomarkers and little clinical translation of neuroimaging» and "this knowledge is necessary to understand atypical neurodevelopment and develop targeted biomarkers and treatments». Is the motivation and goal based in personalized mental health medicine, and that MRI based cortical thickness in the future can be used as a diagnostic tool at the individual level as well as for personalized mental health treatment? This is of course up to the authors, but it could be an idea to tone down such statements and perhaps focus more on the highly valid motivation from the abstract (and end of intro?): "Integrating multilevel brain atlases with normative modeling and population neuroimaging provides a biologically meaningful path to understand typical and atypical brain development in living humans.»

7. Results 2.1. It is stated that "we report on how these multilevel neurobiological markers colocalize and explain CT change patterns between 5 and 90 years of age». Similarly, to my previous comment, is there not an issue with applying atlases based on adult biology on child and adolescent morphology?

8. Results 2.1 I struggled to find what the 148-region division was based on. I first searched here "(CT; Fig. 1B; Text S1.2; age distribution: Fig. S5; CT trajectories: Fig. S6A and Animation S1)", but in the end found a reference to Destrieux within the SI Figure 1 legend, and also later on page 18 under Methods but not stated clearly. Why reduce cortical thickness data based on the Destrieux atlas? This atlas is not based on any of the main candidate mechanisms underlying cortical thickness but instead on larger folding patterns and some anatomical landmarks. As the logic of the study is to use sophisticated cortical divisions, this is not clear to me. A vertex-wise approach would preserve granularity, but then one could not use the open access cortical thickness normative models which I believe are only available for Destrieux and Desikan. I would state and discuss this reduction choice clearly within the text.

9. Results 2.1 Similarly, all the cellular and molecular atlases were first reduced to factors and then mapped to Destrieux if I understand correctly. I do not know the initial granularity or spatial distribution of the multilevel atlases, but could reducing them to 74 "folding based" ROIs per hemisphere be problematic? Great with some clarification here. Also, could this mean that your findings are only relevant for studies that use the Destrieux division? I believe both, ROI-based Desikan- or even vertex-wise approaches are more common than using Destrieux.

10. 2.1. Mapping neurobiological markers to cortical development: Is the cortical microstructure atlas incorrectly referenced? nr 29 (Hansen, J. Y. et al. Mapping neurotransmitter systems to the structural and functional organization of the human neocortex) Is this not the multimodal Glasser atlas? If so, I also believe the correct terms are "T1w/T2w" specifying that this is based on weighted imaging and not (T1/T2) i.e. quantitative relaxometry.

11. 2.1. Mapping neurobiological markers to cortical development: As the Rutherford normative models are based on cross-sectional data I would not call these trajectories cortical "development" or even "change", as they are based on cortical differences at different ages or? I would be

consistent in the terminology and only use development and change when talking about longitudinal data like ABCD/Imagen.

12. Discussion: It was slightly unclear to me whether your findings match the previous main candidate mechanisms for cortical development. To my understanding re-organization of dendritic arbour, cortical myelination, and even synaptic pruning (although controversial as MRI studies cannot be solely explained by small-scale changes at the synapse level) are candidates. Your discussion instead highlight microglia and thus immune responses and dopamine receptor activity is that correct? When I read the discussion, it currently appears as if all your results fit previous literature with no discrepancies discussed. Also, could the relation to the dopamine system be considered controversial just in regard to the mm scale of MRI?

13. Discussion: It is stated that "As promising candidates for clinical translation, we identify the dopaminergic system and microglial cell populations for early development". Would one not have to apply the normative models to clinical populations like in the Rutherford paper, in order to probe clinical relevance? It is unclear to me why non-typical age trajectories should relate to mental health any more than to any other variable that shows stronger associations to imaging and cortical thickness specifically. Like a previous comment, I find the link to clinical utility inconsistent, and in my opinion not necessary for the relevance of the paper.

14. Methods: It is stated that "Given that the main goal of the current analysis was to establish the feasibility of capturing associations between CT development and multilevel neurobiological markers on the individual level, clarification of the sources of .. site-effects will be a task for future investigations. When using multi-site initiatives like ABCD (21 sites, 29 scanners), harmonizing or somehow minimizing the effects of scanner is basic convention. It is not clear why the authors find this a task for the future as the large effects of scanner is already widely documented also specifically within ABCD. The normative models used within the paper attempt to minimize effects of scanner, should the same not pertain to the target longitudinal data? Please explain why scanner bias is not relevant for your analyses or add it as a limitation.

15. Methods It is stated that site effects were estimated in «healthy subsamples» of both dataset's baseline data (n = 20 per site, 50% female) how was healthy defined?

Reviewer #3 (Remarks to the Author):

I previously reviewed this paper for Nature. I have not retained a copy of the version submitted to Nature, and the authors have not provided a rebuttal letter detailing their response to reviewer comments on that submission, so it is difficult to be sure if/how this version has been revised compared to the version previously reviewed for Nature. I think some of my points have been minimally addressed in Discussion, however there don't appear to have been many, if any, substantive changes. Most of my prior comments therefore still stand, as copied below:

"This paper reports the results of an extensive set of analyses, using multiple large prior datasets, conducted by an expert and experienced team using careful and generally well-described methods. Overall the text and figures are composed to a very high standard.

The study investigates spatial colocation of cortical thickness (CT) with multiple molecular imaging maps (referred to as nuclear imaging) and maps of neurotransmitter density and cellular density derived from postmortem data. The basic concept of colocalising MRI phenotypes with genomic, molecular and cellular reference atlases in an effort to understand more about the neurobiological substrates of MRI phenotypes is not novel. This strategy has been widely used in recent years and it has generated interesting results, as in this paper, but it also has well-known limitations. It is clear that the authors are aware of all this. However, I am not sure how decisively this work advances the field beyond what has already been done, or tackles some of the fundamental limitations of spatial colocation analysis for mechanistic interpretation of MRI phenotypes.

The MRI CT maps are derived from normative modelling of ~50,000 scans as previously reported by Rutherford et al, and from the ABCD and IMAGEN datasets, to represent the cortical distribution of thickness as a function of age. Each age-specific CT map is then assessed for spatial co-location with factors derived from 49 prior atlases (Supp Info Table 1). 21 of the atlases are cell-specific gene expression patterns previously derived from the Allen Brain Atlas. Also included is a T1/T2

myelin map from the Human Connectome Project. The remaining 27 atlases are derived from PET data using various transmitter-specific and other radiotracers. The molecular (imaging) and cellular (post mortem) atlases were then separately subject to factor analysis, resulting in 10 molecular and 10 cellular factors + the T1/T2 map = 21 so-called "multi-level brain systems". Each of the 21 factors was then tested for spatial colocation with the CT data cross-sectionally and with CT change scores for 5 year epochs by a sliding window analysis in the age range 5-90 years. I have a number of comments, questions about this approach.

1. The PET atlases are quite diverse in terms of age (mean age ranging from 22-68 years across atlases) and sample size (N ranging from 6-204). The ABA atlases are derived from N=6 with mean age 42.5 year. None of the reference atlases provides data over the same age range as the CT trajectories. The implications of this mis-match are briefly mentioned (as "noise", in the last para of discussion) but not properly addressed. Do the authors assume that the cortical distribution of, say, the dopamine transporter (DAT), is unchanged over the life-cycle, and therefore that the available DAT map (mean age 61 years) is a reliable indicator of DAT distribution in adolescence? This assumption seems unlikely to be true, either for DAT or in general across all the reference atlases; but without it how we can make any mechanistic interpretation of colocation between developmentally varying CT measurements and fixed-in-time reference atlases? As far as I can see, the authors' principal defence on this point is that some of their results are consistent with what is already known about the developmental dynamics of some molecules or cells in the brain. For example, in the Discussion, they note that the strong colocation of the dopamine-based factor with CT change in adolescence is consistent with prior data showing that D1 receptor density peaks during adolescence. Fig 6 descriptively summarises the context for some of the results reported in this paper compared to prior reports of developmental change in other molecules or cells. Personally, I didn't find this very convincing. I understand that there are limitations to the data available for use as reference atlases for lifecycle development – but despite identifying this as a problematic aspect of prior studies using the ABA dataset (in Introduction), and their brief comments in Discussion, the authors could do more to convince the reader that spatial co-location of two maps measured at very different points in the lifecycle is mechanistically relevant (rather than descriptively interesting) for deeper understanding of brain developmental processes. I suggest this is particularly important because the analysis is correlational not causal, and the causal mechanistic significance of co-located "multi-level brain systems" and MRI maps would be debatable if the MLBS and MRI data were measured in demographically matched cohorts, or even from combined PET/MRI data in the same cohort, e.g., do dopamine receptor changes in adolescence drive CT changes, or vice versa?

2. There is insufficient detail provided about some aspects of the factor analysis used to generate "multi-level brain systems". The following points were not clear to me: How were the reference atlases differentially weighted on each factor? How important was each of the 10 cell (or molecular) factors in terms of the proportion of total variance in the "raw" atlases that they accounted for? What was the extent of correlation between MLBS's derived from non-orthogonal factor analysis? Without these data it is very difficult for a reader to make up their own mind about interpretation of the results of correlating these derived factors with CT maps. The rationale for progressively stepping down from 21 MLBS's to 9 and then 6 and then focusing on a few "raw" PET atlases, over the course of the results, could also have been more clearly signposted in advance or justified.

3. It is commendable that the authors endeavour to generalise results derived from the Rutherford et al MRI dataset to ABCD and IMAGEN datasets including longitudinal data. It is reported that the MLBS co-location results reported on the basis of the Rutherford dataset are somewhat replicated in the ABCD and IMAGEN datasets (R² 25-56%). Since the MLBS's are unchanged for the Rutherford and ABCD/IMAGEN analyses, presumably this degree of correspondence simply reflects the correlation between Rutherford and ABCD/IMAGEN CT maps? Likewise, the high degree of variability in MLBS-coupling noted at an individual level presumably reflects the typicality or atypicality of an individual's CT phenotype compared to the developmental norms published by Rutherford? I apologise if I have missed something in the rather dense description of these analyses but showing that datasets A and B (Rutherford and ABCD/IMAGEN) are both correlated with fixed point of reference C (MLBS) surely only tells us that A and B are correlated? This would not be surprising, especially since the Rutherford dataset includes (baseline, cross-sectional) data

from the ABCD cohort, i.e., A and B are not entirely independent of each other by construction. I would welcome greater clarity concerning the relationships between the MRI phenotypes (A and B), and the extent to which variability in individual MLBS-coupling reflects individual (a)typicality of MRI phenotypes compared to age-appropriate norms. As it stands, I found it difficult to see how the analysis of ABCD/IMAGEN longitudinal data, in aggregate or individually, added materially to the robustness or biological interpretation of the principal results derived from the Rutherford dataset.

4. The Discussion suggests that these methods and results can be used to validate new treatment targets, and/or to understand the molecular or cellular basis of atypical MRI phenotypes in neurodevelopmental disorders. However, there are no data reported on disorders and the only "targets" suggested by these results are "dopaminergic and microglial systems for early development as well as the cholinergic system in context of pathological aging". This level of target validation is too broad-brush for practical purposes, e.g., drug development, and I'm not convinced it tells us anything we didn't already know. Likewise, I think the claim (p15) that "spatial colocalization approaches can facilitate discovery of physiological mechanisms underlying specific conditions" is not really nailed down by the data that have been reported, due to the limitations of spatial correlational analysis for clarifying mechanistic relationships between correlated maps."

Dear Reviewers,

We appreciate the time and effort dedicated to evaluating our manuscript, and we are grateful for the insightful and constructive feedback provided. We have incorporated all comments into an extensively revised version of our manuscript, which we think has now been substantially improved. In our responses, we first address the issue that every reviewer raised: whether it is generally sensible to conduct spatial correlation analyses between child/adolescent CT and adult neurobiological atlases; followed by specific responses to each reviewer’s statements. All original reviewer comments are printed in **blue font**, citations from the revised manuscript are printed in **red**. We hope that our responses and adjustments sufficiently address all points raised by the reviewers.

Sincerely,

Leon D. Lotter and Juergen Dukart, on behalf of the authors

Spatial correlation analyses between developmental CT and adult neurobiological markers:

We chose to discuss this topic upfront as it was mentioned by every reviewer separately. In discussion of the individual comments below, we will point back to the current section where appropriate.

The reviewers raised important points regarding the sensibility and validity of spatial association analyses between pediatric or adolescent MRI data on the one side (here: CT change) and adult neurobiological templates on the other (here: mainly PET and gene expression maps). Our response is split into three sections, each followed by changes made to the manuscript: (i) the mechanistic assumption underlying our analysis framework, (ii) the question if our framework implies causal relationships, and (iii) the inevitability of spatial association analyses if we want to comprehensively study biological correlates of human brain development,

Mechanistic assumptions:

As laid out briefly in the manuscript, our framework builds on the following reasoning: Cortical thickness changes over the lifetime, from pre-birth to late adulthood. These changes do not appear uniformly across the cortex, but follow distinct trajectories by cortex region. For example, while two cortex regions might show a comparable overall decline in CT when evaluated

across the whole lifespan, they might simultaneously show distinct trajectories, with individual peaks and plateau phases at different timepoints during the lifespan (c.f. examples in Fig. 1B). Evaluated across the whole cortex, this will result in a specific cortical pattern associated with any given developmental period (c.f., Fig. S6B/C). We assume that these patterns are not random but reflect biological processes that causally influence CT changes over time. A classic example would be the regionally-specific development of synaptic density and dendritic arbor as general markers of neuronal change, which might underlie developmental trajectories of macroscale brain structure. Unfortunately, much of the evidence for these relationships originates from research done decades ago, most noteworthy the works by Huttenlocher and colleagues (Huttenlocher, 1979, 1990; Huttenlocher & Dabholkar, 1997) and later Petanjek *et al.* (Petanjek *et al.*, 2008, 2011). Major further evidence was then only provided by Paus *et al.*'s spatial correlation studies based on Allen Brain Atlas data (Parker *et al.*, 2020; Patel *et al.*, 2019; Shin *et al.*, 2018; Vidal-Pineiro *et al.*, 2020; Wong *et al.*, 2018; Paus, 2018). In line with Paus and colleagues, we work with the following premises:

- 1) **X** is a neurobiological entity (e.g., dendrites, specific neurons, or neurotransmitter receptors) that changes with human neurodevelopment and aging.
- 2) Changes in **X** might have downstream effects on CT as measured with MRI, either directly (e.g., more/less neurons or dendrites lead to higher/lower CT) or indirectly (e.g., higher density of a neurotransmitter receptor as a proxy of certain cellular structures developing/degrading; cortical myelination leading to changes in gray-white-matter contrast). Alternatively, there might be a common neurobiological process **Y**, which independently affects both **X** and CT, leading to a correlation between changes in **X** and changes in CT that does not require a direct causal relationship between **X** and CT but would still imply a common underlying mechanism shaping both.
- 3) **X** is in appearance and strength distributed non-homogeneously across cortical regions, resulting in a (more or less) specific spatial distribution.
- 4) The cross-regional distribution of **X** is not necessarily stable across the human lifespan, but it might vary with neurodevelopment and aging. However, a major change of **X**'s distribution would require that multiple brain regions change their relative ranks of **X** density/expression/activity as compared to, e.g., a general decrease of **X** in all regions or a marginal increase of **X** in one region that does not lead to a relevant “region rank-increase”.

Such a major reorganization process is relatively more likely during neurodevelopment (i.e., pre-birth to maximally early adulthood) and aging (late adulthood) than during the relatively stable middle adulthood period.

- 5) If changes in **X** or **Y** led to changes in CT, the spatial distribution of these CT changes would likely resemble the “steady-state” adult spatial distribution of **X/Y**. This can be assumed as the stable adult distribution of a given neurobiological entity has to be the result of its neurodevelopment, and also the beginning-point of its aging processes. Therefore, changes in **X/Y** that lead to CT changes will likely be strongest in cortex regions with high density/expression/activity in **X/Y**’s “steady state” as this is either the state that they are approaching (neurodevelopmental CT changes) or coming from (aging CT changes).

From these assumptions follows that an observed spatial correlation between (i) the distribution of **X** as measured during the stable period and (ii) the distribution of CT changes during a given developmental period could have resulted from a developmental process that **X** is subject to. Noteworthy, this by no means is the only explanation for such CT changes, which is discussed in the next section on the topic of causality. Furthermore, although we consider them conceptually sound, it is currently not possible to prove all of our assumptions and specific research on the matter – especially on cross-regional spatial patterns of (developmental) neurobiological processes – is extremely scarce. Finally, the described framework assumes an idealized scenario in which neither measures of CT change nor of **X** are subject to noise and in which all “neurobiological markers” of **X** are measured during the stable mid-adulthood period.

As this topic was raised by all reviewers to some degree, we took great care in revising our explanation of the theoretical argument underlying our approach as outlined in the Introduction (third Introduction paragraph, p. 2) and critically evaluated in the Discussion (third Discussion paragraph, p. 12). In the latter critical evaluation, we explicitly refer to the conceptual and methodological issues to which our approach might be subject, as well as to the interpretive caveats discussed in the next section on causality. In this context, we have distributed the final “limitations” paragraph across the discussion in order to address each topic in the context of the results and thus emphasize their relevance. A point raised by two reviewers pertained to premise (4) on the stability of cross-region ranks of neurobiological markers during mid-adulthood. While this obviously could be shown empirically, data to do so for all of the evaluated neurobiological markers is scarce and would require gene expression and nuclear imaging assessments in large samples, across a broad

age range, and with high spatial resolution. To address this issue as far as possible with existing data, we obtained additional nuclear imaging maps that were not included in our main analyses but used the same tracer in other age groups and showed that, for this small set of targets/tracers (mGluR5, D2, VACht), cross-region ranks are indeed stable during adulthood (Fig. S2). Indeed, this analysis was already included in the version of the manuscript submitted to *Nature Communications*, in response to the reviewer comments we received after submitting to *Nature* (c.f. Reviewer #3). However, the results were not emphasized in the main manuscript, which we now addressed in Results section 2.1 (“In support of our analytical rationale, three neurotransmitter receptors/transporters, for which alternative atlases from different adult age groups were available, showed high stability of spatial patterns during adulthood (Spearman’s $\rho \geq 0.68$; Text S1.2.3, Fig. S2).”). Relatedly, we attempted to provide a low-level confirmation of our spatial association results by assessing developmental trajectories of the studied neurobiological markers directly in developmental gene expression data (Kang et al., 2011) (Fig. 4). As above, this analysis was added in response to the *Nature* reviewers’ requests for validation of our results (c.f. Reviewer #3). While we presented the results in context with our main CT change association findings, they also carry some informative value in support of the underlying assumptions for spatial correlation analyses, as stated above. We now highlight this explicitly in the Results (Section 2.8, “Conversely, we observed relatively stable phases during mid-adulthood for most genes/gene sets, supporting our mechanistic rationale (see Introduction).”).

Causality:

As elaborated thoroughly above, our approach makes some causal assumptions about possible relationships between neurobiological entities and CT. Such assumptions, however, somewhat underlie by far the most neuroimaging studies and inferential statistical analyses, as the goal of such studies and analyses is to test for specific hypotheses to approach the underlying, ideally causal relationships, between specific variables. We explicitly do not argue that our results concerning spatial associations between CT changes and neurobiological markers anyhow “prove” causal relationships. At most, we consider them as indicators of a potential mechanistic relationship between both as per the above arguments. Our study design, as basically any study design without a causal intervention, does not have the ability to confirm causal relationships. Furthermore, even if assuming some causality in a spatial relationship we report on, the specificity of such an association would be limited by many methodological issues such as the (i) the spatial resolution

of CT change data, i.e., the applied Destrieux 148-region parcellation, (ii) the spatial resolution of neurobiological markers, i.e., the resolution of nuclear imaging and the sampling of gene expression data, and (iii) intercorrelated spatial distributions of neurobiological markers.

Considering all of these limitations, we took great care in the revision of our manuscript to (i) correctly contextualize the description of our premises, our methodology, and our results, especially noting their inability to prove causality (third Introduction paragraph, p. 2: “**We emphasize that, despite causal assumptions being made on the conceptual side (see above), neither prior nor the current spatial correlation study can actually *prove* causal relationships between an MRI-observed change pattern and tested neurobiological markers. Relatedly, the specificity of spatial associations is inherently limited by the spatial resolution and noise associated with both correlated components.**”; Supplementary Text S1.1), and (ii) added a new paragraph to the discussion in which we dive into the issue of causality and specificity in spatial correlation analyses and outline potential research paths to evolve the approach further (third Discussion paragraph, p. 12).

The argument for spatial correlation analyses (with “normative” neurobiological markers):

We clearly note that spatial correlation approaches would be rather unnecessary if we could measure a given biological entity in humans in-vivo and at scale. However, this is only hardly, or often not at all, possible given today’s neuroimaging tools. Activity or structural changes of neuronal or glial cells cannot be measured in-vivo in healthy/typical developing humans and post-mortem data, especially considering earlier neurodevelopmental periods, is extremely scarce and may suffer from other underlying biases. Today, neurotransmitter activity, brain immune activity, and, to a lesser degree, brain metabolism can be measured to a very limited degree in-vivo in humans only using nuclear imaging techniques. The use of such techniques for study of brain development, however, is subject to severe practical and ethical challenges. Aside from the practical issues (i.e., high effort to maintain a nuclear imaging facility and associated high costs, reducing feasible sample sizes), the exposure to radioactivity practically forbids application in healthy children and adolescents. Given these strong methodological and ethical limitations, to date the spatial correlation approach is the most reasonable route to take. In the context of the methodological assumptions discussed above, the reviewers suggested using age-specific templates of “neurobiological markers” in our analyses. As lined out above, our argument is indeed based on the idea of using “fixed” markers obtained during a relatively stable human developmental

period as a reference. Use of age-adjusted atlases would basically prevent the interpretation of correlational findings based on the logic discussed above, i.e. a change in correlation could be attributed to a change in the atlas and/or the functional data, or alternatively, if both are mechanistically linked, a simultaneous change in both measures could result in a stable correlation. We nevertheless agree that, using a modified analysis framework, much information could be gained from using age-specific templates. However, due to the constraints discussed above, we do not believe that this is possible in the foreseeable future.

As requested by Reviewer #2, we have also extended on the discussion of “practical obstacles” limiting the availability of human data, and on the resulting argument for spatial correlation analyses, in the Introduction (second paragraph, p. 1–2).

We hope that this detailed argumentation provides a compelling case for the use of spatial correlation analyses as a viable tool to approach the mechanisms underlying life-time CT changes as evaluated in our study. In the following, we address in a point-by-point reply the more specific comments raised by each reviewer.

Reviewer #1:

In this manuscript, the authors investigated spatial associations between many neurobiological signatures and lifespan changes in cortical morphology. Specifically, they collected 49 postmortem and in vivo brain atlases covering molecular and cellular processes in healthy adult samples, which were linked to lifespan changes in cortical thickness. This is a timely and important study, revealing a potential neurobiological basis for the maturation of cortical morphology. Here, I have several major and minor concerns, which are outlined below. I hope that these comments and suggestions can be used to further improve the manuscript prior to publication.

We thank reviewer #1 for their positive evaluation of our work! We hope that our changes to the manuscript along with our responses provided below sufficiently address their concerns.

The authors hypothesized that spatiotemporal patterns of thickness development colocalize with nonpathological adult spatial distributions of the respective neurobiological markers. Given that many neurobiological signatures, such as brain metabolism, develop with increasing age, the

authors should clearly address and validate the validity of spatial association analysis between lifespan changes in thickness and adult neurobiological markers.

We thank the reviewer for this suggestion that was also raised by both other reviewers. We have now discussed this issue in detail at the beginning of this response letter and integrated the respective changes into the revised manuscript. We hope that we have now sufficiently addressed the concerns of the reviewer in this regard.

Title: This study is a spatial association analysis and does not address how human cortex development is shaped by molecular and cellular brain systems.

In line with our more careful and extended treatment of the topic of causality as discussed in the introductory part of this response letter, we changed the title to “**Regional patterns of human cortex development colocalize with underlying neurobiology**”. We consider this choice to be more informative about the correlational nature of our study.

Introduction. The manuscript does not provide sufficient detail on the potential mechanisms by which these molecular and cellular processes influence the development of cortical thickness. A further description would be beneficial.

We agree with the reviewer and now elaborate in more detail on potential mechanisms through which the investigated spatial relationships could arise (second Introduction paragraph): “Given the multitude of neurobiological mechanisms that likely shape cortex morphology over the lifetime, it is to assume that CT change patterns at any given developmental period result from several interacting biological factors jointly influencing cortical microstructure as outlined above. For example, concerted developments across cortical cell populations could be mediated via specific neurotransmitter effects projected from deeper brain regions, as was indicated in early non-human animal studies for, e.g., glutamatergic and serotonergic effects of thalamocortical projections on motor and somatosensory cortices as well as for dopaminergic effects of mesocortical projections on the prefrontal cortex. Relatedly, neurotransmitter receptors likely play regulatory roles in cortical development as evidenced, for instance, by the effects of in-utero cocaine exposure on cortical macrostructure, thought to be caused by a disruption D1 and D2 dopaminergic receptor influences during cortex development.”

The brainchart data are from a previously published study (Rutherford et al.). However, from the age distribution shown in Fig. S5, there are very different numbers of subjects at different ages. In

particular, there are very few subjects before age 10 and around 40. Is the brainchart data affected by these confounding variables?

We thank the reviewer for this comment on a potential confound. The age distribution (Fig. S5) resembles a pattern found in most lifespan normative modeling studies published to date (e.g., Bethlehem *et al.*, 2022). The pattern is caused by comparatively large groups of subjects from the ABCD and UK-Biobank studies. However, we would like to note that the conclusion of only “very few” subjects before 10 and after 40 years is biased by the visualization due to the over-proportionally high number in the other age range. Specifically, from 5 years on, the sample consistently contains over 50 subjects per year, which is in line with the current estimate of about 3,000 subjects necessary for robust lifespan modeling of brain data provided by the ENIGMA Lifespan Working Group (Ge *et al.*, 2024). We nevertheless assessed the potential influences of the non-uniform age distribution on our first main outcome, i.e., the CT change variance explained in the individual, univariate regression models (c.f., Fig. 3, lower row). Figure S12 shows the correlations between (i) the number of Braincharts subjects per age-window in our main analyses (5-year length, 1-year steps) and (ii) the R² values indicating univariate and multivariate associations between CT change and neurobiological markers. We assessed significance of these associations based on the marker null maps used in our main analyses and found that none of the markers which explained CT changes (FDR-corrected) showed a significant correlation to the age distribution indicating that these findings are not induced by this confound (Supplementary Text S1.3.3). However, as such an analysis may not be sufficient to fully exclude any potential confounds induced by the non-uniform Braincharts age distribution we have also added this limitation at the appropriate place in the Discussion (second paragraph; p. 11).

Lines 1–10, Page 6. The authors show the spatial associations between cross-sectional thickness and neurobiological features and describe diverse colocalization trajectories. However, the results are not clearly described. The authors found a general pattern of strongest changes from childhood to young adulthood (up to approximately 30 years) as well as in late adulthood (from 60 years onwards; Fig. S7). A possible reason for this result is that the neurobiological maps are from adult brains. Figure S7 should be moved to the main text. The authors stated that sex did not relevantly influence the trajectories (Fig. 8). Did they perform statistical comparisons for sex differences?

We thank the reviewer for their suggestion to put stronger emphasis on the “cross-sectional” spatial colocalization patterns between modeled CT and neurobiological markers. The respective

analysis rather served demonstrational purposes as the core of the paper (as emphasized throughout the manuscript and now also in considerably more detail in the introduction) is on the relationship between changes of CT and potentially underlying neurobiology. For this reason, the prior Fig. S7 was not included in the main manuscript, and we chose to not describe individual cross-sectional trajectories in detail. However, following the reviewer's comment, we have now moved the Figure to the main manuscript (now Fig. 2, moving prior Figures 2–6 to position 3–7). The Figure shows the trajectories of (Spearman) colocalization estimates between the 21 neurobiological markers and cross-sectional modeled CT as extracted for year 5 to 90 from the normative model. To descriptively support our claim of “the strongest changes up until 30 years and from 60 years onwards”, we added change plots, visualizing, for every timepoint, the change in colocalization across the next 1, 2, 3, 4, and 5 years. In response to a following comment by this reviewer, we also included left and right hemisphere surface projections of the neurobiological markers in this figure in order to display them in the main manuscript. As these colocalization analyses of cross-sectional modeled CT (also pertaining to the analyses of later modeled CT change) are based on predictions from the normative model, we cannot perform valid statistical tests for potential sex differences but only describe them visually. To facilitate this, we added a new Figure (S8), showing the LOESS curves fitted to the predicted CT colocalization data for each sex in direct comparison. Here, we see variable differences in colocalization strength, but no obvious differences in trajectories. We now refer to this also in the results description (Section 2.3: “**On visual comparison, trajectories appeared similar across sexes but partly differed in overall colocalization strength (Fig. S8). The modeled nature of the colocalization estimates precluded statistical tests, which would need to be conducted in individual-level data.**”). Finally, regarding the potential impact of neurobiological marker age on modeled CT colocalization, we added an additional sensitivity analysis similar to the analysis on potential associations between the Braincharts age distribution and explained modeled CT change described in the preceding comment. In Supplementary Text S1.3.3 and Fig. S13, we test if the explained modeled CT change was highest in time windows most distant from the respective “marker source age”. We are aware that the reviewer mentioned this concern in context with the cross-sectional colocalization results but as the (pseudo-)longitudinal results are the focus of this work, we decided to perform all major sensitivity analyses on the modeled CT change data. We do not find evidence for such systematic relationships. Rather the peaks of explained modeled CT change were mostly located before or

close to the respective marker source age (ages distributed around 30 to 50 years), in line with the notion that cortex development continues into early adulthood.

The authors showed the spatial associations between neurobiological markers and changes in cortical morphology. Several studies have demonstrated a network mechanism underlying the maturation of cortical morphology (e.g., PMID: 27457931; PMID: 38278807). I would suggest adding a discussion on this topic.

We agree with the reviewer that this is a reasonable route to explore in context with our results. We now cite the provided studies in the discussion, introducing them from the perspective of diverging development of unimodal and transmodal cortical areas. While, based on the residual difference (i.e., “leave-one-region-out”) analyses, we find that some cortical areas typically implicated in such studies seem to be relevant for our results, the overall patterns do not match. E.g., oftentimes the cingulate or parahippocampal cortex is implicated in a spatial colocalization together with motor or unimodal medial occipital cortices – a pattern that does not fit in the unimodal/transmodal literature and surely requires further investigation (first Discussion paragraph, p. 11).

The curves presented in Fig. 2 appear to differ from those in Fig. 3A. It is unclear which result should be referenced to understand which molecular and cellular markers more significantly constrain cortical thickness development at specific ages. Please clarify.

We apologize that we did not sufficiently highlight our hierarchical analysis workflow enough throughout the paper. We now emphasize our analysis workflow more explicitly in the second subsection of the Results, along the structure provided with Fig. 1. Furthermore, during the reporting of actual results, we clearly describe the applied hierarchical procedure. Specifically, Fig. 3 (previously Fig. 2) shows how the 21 neurobiological markers jointly (multivariate regression, upper panel) or separately (univariate regression, lower panel) explain modeled CT change. As described in the manuscript, Figure 4 (previously Fig. 3) shows dominance analysis results for the 9 biological markers that significantly explained modeled CT change (FDR-corrected) at any time window in the univariate analyses. Dominance analyses can quantify the contribution to the overall R^2 in a multivariate linear regression. Therefore, as the reviewer mentioned, the curves differ between the two analyses as the first one looks at all predictors independently and the second one looks at the variance explained by a subset of these predictors, while accounting for shared variance between predictors. In addition to a brief description in the Results section (Section 2.5, “Using

dominance analyses to quantify the individual contribution of each univariately FDR-significant neurobiological marker in a multivariate setting [...]”) and the detailed explanation in the Methods (section 4.5.3), we now added additional labels to Figure 3 and 4 (left side) to highlight each analysis methodology.

Line 21, Page 2. Introduction. The authors describe that "neurodevelopmental disorders are associated with both deviations in brain structure and dysfunction of several neurotransmitter systems, but suffer from a lack of reproducible biomarkers and little clinical translation of neuroimaging research.". Many previous studies have reported that neurodevelopmental disorders such as autism and depression are associated with deviations not only in brain structure but also in functional connectivity. Moreover, the reproducibility of results has been well studied using multisite neuroimaging data. The authors need to rephrase their claims. In addition, this paragraph describes the importance of neurotransmitters and the normative model in understanding disease or atypical development. However, this is not reported in the results section, which is confusing.

We apologize for this misleading statement. When referring to the lack of reproducibility of biomarkers, we were referring to the lack of actually clinically meaningful/applicable biomarkers that are usable for individual diagnosis or monitoring. The reviewer is absolutely correct that there are structural and functional neuroimaging biomarkers for various neurodevelopmental disorders that were consistently replicated on a group level across different studies and cohorts. However, to our knowledge, all of the respective studies demonstrated rather weak to moderate effect sizes that up to now did not allow for translation of the respective measures into the clinical routine. Due to removal of several references to diseases and atypical development to avoid misleading the reader regarding the scope of this study, the statement to which the reviewer refers was dropped from the manuscript.

Line 5, Page 3. Introduction. The authors described that "multimodal neuroimaging-based spatial association approaches can provide a window into specific biological mechanisms, but until now were limited to postmortem data and the cellular level. " In fact, there are many multimodal neuroimaging studies showing spatial associations between brain signatures such as brain connectivity and metabolism (PMID: 23319644; PMID: 33443160; PMID: 30741935).

We agree with the reviewer and apologize for this imprecise statement. In fact, the sentence was directed at developmental studies and was corrected to: “**Multimodal neuroimaging-based**

spatial colocalization approaches can provide a window into specific biological mechanisms, but – to our knowledge – developmental studies until now were limited to *postmortem* data.”.

Line 30, Page 3. 21 neurobiological maps should be presented in the main text, given their importance for the association analysis. Results: Ni8 was not mentioned.

We thank the reviewer for the suggestion to show plots of each neurobiological marker in full in the main text. We would like to highlight that this was already done in a limited format in the original manuscript, namely on the right side of (now) Figure 4. As mentioned in another comment above, we now include surface projections of the neurobiological markers in the cross-sectional colocalization plots (Fig. 2) to provide an overview to the readers. Additionally, we keep the full plot of the original (Fig. S1) and dimensionality-reduced (Fig. S4) neurobiological markers in the supplements. Regarding the results reporting, given the multitude of candidate marker studies and formatting requirements, to facilitate readability, we limited specific mentioning in the results to only those that significantly explained (modeled) CT development. Ni8, along with some other markers, did not pass this criterion.

Line 18, Page 6. The authors showed that molecular- or cellular-level markers explained up to 54% of the spatial variance. How about the combined molecular and cellular-level markers? How about the sex differences?

All 21 neurobiological markers combined explained up to 67% of modeled CT changes, peaking during the adolescence-to-adulthood transition (Fig S9). These information and analysis were indeed already included in the original manuscript, after the univariate results. We agree that this might not have been the logical order of results reporting and now moved it to directly after the cellular/molecular marker set-wise results (“54%”, as the reviewer refers to above), but before the univariate results.

Line 8, Page 8. In the main text, the description of Fig. 3A states, "We found that the nine FDR-significant molecular and cellular markers jointly explained 58% of CT change patterns from 5 to 30 years." However, Fig. 3A is labeled as "neurobiology markers significant in univariate analyses." This raises the question of whether the analysis is multivariate or univariate. Please clarify.

We apologize for this imprecise labeling of the figures. We adjusted the heading of Fig. 4A (prior Fig. 3A) to “Selected neurobiological markers”, as the dominance analyses reported here

were performed on only those markers that turned out FDR-corrected significant in the univariate analyses (Fig. 3, prior Fig. 2). In addition, we adjusted the figure legend and added additional labels to Fig. 3 and Fig. 4 informing the reader about the types of analyses performed (Fig. 3: “univariate regression”, “multivariate regression”; Fig. 4: “dominance analysis”, “Spearman correlation”).

In Fig. 6, the meaning of the colored overlay is unclear. What do different shades of the same color represent (for example, in the context of synaptic pruning) (Huttenlocher & Dabholkar, 1997)?

Different shades of the same color in Fig. 7 (prior Fig. 6) are indeed bars that were laid over each other (appearing as different shades). As described in the legend, these highlight cases when a study reported results for similar processes but for, e.g., different brain regions (e.g., Huttenlocher & Dabholkar, 1997, report synaptic pruning for multiple brain regions). Details (listings for, e.g., different brain regions) can be assessed in Tab. S5.

The discussion section lacks an in-depth exploration of the physiological significance reflected by the constraint of relevant neurobiological markers on cortical maturation at specific developmental stages. This should be a central concern of the paper and needs further elaboration.

In response to this and other reviewer remarks, we extended the introduction of the cellular-level mechanisms that might lead to macroscale CT changes (e.g., neuronal/glia reorganization incl. remodeling of dendritic arbor, pericortical myelination) significantly (first Introduction paragraph, p. 1). We then introduce potential pathways through which alterations in neurotransmitter systems could influence the cellular-level determinants of CT (second Introduction paragraph, p. 1). In comparison to the original manuscript, the first paragraph of the Discussion on convergence between our spatial colocalization findings and prior literature was expanded (p. 10). However, we argue that a much more detailed treatment of potential specific mechanisms that could underlie our associative findings would not only go beyond the scope of this work but might also be rather speculative given the non-causal nature of spatial colocalization analyses (see introductory section of response letter). Instead, in a new paragraph, we now explicitly speak towards the fact that, while many of our results converge with the literature, we cannot finally confirm our results or provide explanations for non-converging findings (second Discussion paragraph, p. 11). Here, we also directly discuss the limitations of our approach, which in part also limit specific discussion. In the subsequent discussion, we go into detail on how the approach could be evolved further, on one side, to provide better evidence for potential causal relationships between cortex development and the studied neurobiological processes/entities and,

on the other side, to explore its value from a clinical viewpoint (third Discussion paragraph, p. 12). We think that this focus of the discussion (and introduction) is best suited to our results and hope that the changes are in line with the reviewer’s point of view.

Reviewer #2:

The study integrates numerous “multilevel” cortical atlases, normative trajectories of MRI based cortical thickness and partly separate population-based longitudinal youth samples in an attempt to improve our understanding of the neurobiology underlying cortical thinning across the life span. I find the study comprehensive and highly relevant for a broad field within MRI based neuroimaging i.e. all studies assessing cortical macrostructure.

My comments are addressed below in chronological order and I sincerely hope they can be of some help. As this is a multidisciplinary study, I would like to state that I am a neuroscientist and refer to other reviewers to better scrutinize topics outside of neuroimaging. Also, please do not hesitate to correct me whenever I have misunderstood something!

We thank the reviewer for the overall positive feedback on our manuscript. We hope to have answered their comments and concerns sufficiently in the following responses and revisions made to our manuscript.

1. Introduction: I believe reference nr. 4 regarding the link between MRI based cortical development and cognitive development from the year 2000 is dated and could be updated.

We agree with this notion and replaced this introductory reference with two references on a newer study on relationships between brain development and cognition in the Philadelphia Neuroimaging Cohort (Erus et al., 2015) and a very recent viewpoint on how human-specific neurodevelopmental mechanisms on cellular and molecular levels could support human cognitive abilities (Vanderhaeghen & Polleux, 2023).

2. Introduction: I find the use of the terms “development” vs “change” somewhat unclear, particularly when attempting to grasp the underlying logic of the paper i.e: “Assuming that CT changes over the lifespan are shaped by activity, development, or degeneration of cell populations and molecular processes, it is to be hypothesized that the spatiotemporal patterns of CT development colocalize with nonpathological adult spatial distributions of the respective

neurobiological marker. Increased colocalization of CT changes with an individual marker at a given developmental period would then support a role of the associated cell population or process in respective CT changes.» For the most part it appears that development is not simply “change over time”, which then could pertain to the full lifespan but instead more closely relates to “maturational” processes found in youth only, is that correct? And that for the adult part of the lifespan the word “change” is often used to capture a broader set of processes? I think different disciplines use the word “development” quite differently, so maybe either define in a word or two or be very concise about usage.

We appreciate the reviewer’s comment on this issue and understand that the terminology might have been confusing at times. The sentence to which the reviewer refers to in their comment was removed from the paper and replaced by a more detailed section as elaborated below and especially in the first separate part of this response letter. We furthermore took care to revise the manuscript with a clearer terminology: We now use “development” and “aging” when we talk about (mostly) CT changes during the respective lifespan time windows (i.e., until about 30 years as “development” and in later life about “aging”), e.g., in the abstract: **“We demonstrate that human cerebral cortex development and aging trajectories unfold along patterns of molecular and cellular brain organization, traceable from population-level to individual developmental trajectories.”**. The term “change” is used in a more general sense, i.e., write we talk about the whole lifespan, e.g., in the abstract: **“Human brain morphology undergoes complex changes over the lifespan.”**, or when we refer to timepoint-to-timepoint CT changes during results reporting, e.g., **“In contrast, modeled cortical change patterns during adulthood are best explained by cholinergic and glutamatergic neurotransmitter receptor and transporter distributions.”**. As can be seen in the cited section, if we refer to change during a specific lifespan period, this is specified as, e.g., “adult” or “adolescent change”, with definitions of the timespans provided in Fig. 3 and 4. To clearly label all analyses that were based on 50th-percentile normative model predictions, we here now refer to “modeled CT (change)” consistently throughout the paper.

3. Introduction: The example paragraph from comment 2 holds vital logic for the paper but was (to me) a heavy read, where I had to dissect every sentence. Is there a way to, when presenting this very central concept for the first time, do it “better”/simpler?

- I think this text captures why it would make sense to map cortical thinning in youth onto adult biological architecture, but I could not fully grasp why that is not problematic. Particularly as the

introduction previously states that different mechanisms might underlay youth based- and adult thinning.

- The reader could also be guided better through this very first introduction of increased co-localization. Am I correct in that if a marker is predominantly found in a specific region and those regions show either a more pronounced thickness change, or a change that is steady but improves in precision to the underlying distribution (the spatial patterns “match better”), or a degree of change that more closely match the “concentration” of the neurobiological marker, then all these instances would increase co-localization?

- Is the passive “it is to be hypothesized” a general statement or your actual hypothesis? Is there a suggestion of causation in this statement?

We thank the reviewer for the detailed questions on this issue, which helped us in reformulating the manuscript’s sections in question. As this was an issue raised by all reviewers to different degrees, we formulated a joint response statement at the beginning of this letter. All manuscript sections to which the reviewer refers in this comment were rewritten to provide a more detailed and better understandable account of our rationale. With this, we hope to have sufficiently addressed the reviewers' remarks.

4. Introduction: I find the introduction to some of the biological terminology at times unclear. the term “(multilevel) neurobiological markers” is introduced first in regard to cellular densities and functional properties although not used too often within the text. There is also a distinction between molecular and cellular levels which is used much more frequently including within the title, and as a strength of the paper, but molecular is never formally introduced in the same way cellular is. If I understand correctly in this context “molecular brain atlases» relate mostly to neurotransmitters (and the functional level?) and are based on nuclear imaging. Is there a more structured way to present these topics? Moreover, I am wondering if “cell type”/“cell density”/“cell population”, and “process”/“function” is also used interchangeably at times, but I might be wrong here, so great if the authors could have a look, but they know best.

We appreciate the reviewer’s thoughts on this issue. In our general revision of the Introduction, we now introduce and use the terms “molecular” and “cellular” more carefully: The Introduction now leads with the cellular-level mechanisms that might lead to macroscale CT changes (e.g., neuronal/glial reorganization incl. remodeling of dendritic arbor, pericortical

myelination). We then introduce potential pathways through which alterations in neurotransmitter systems could influence the cellular-level determinants of CT as for example through effects of projections from deeper brain structures or regulatory roles of dopamine receptors (second Introduction paragraph, p. 1). Keeping this hierarchical perspective, we distinguish the “cellular” from the “molecular” level, with “molecular” being used as an umbrella term for processes organized on a lower organizational level (across, e.g., cell populations) that are accessible via nuclear imaging. We later introduce “cellular markers” and “molecular markers” specifically as the terms describing the two sets of neurobiological brain maps used in our analyses, the first (“cellular”) referring to gene-expression maps of cell populations and the second (“molecular”) describing nuclear imaging-measured entities that are likely distributed across cell populations, such as neurotransmitter receptors or metabolism. We reckon that with this treatment we make clear that “cellular/molecular marker” is explicitly used as an umbrella term. Relatedly, we dropped the term “multilevel” from the manuscript as it, indeed, did not add greatly to reporting. The maps used as “predictors” of CT changes are now only referred to as “neurobiological (cellular/molecular) markers” throughout the manuscript.

5. Introduction: What are the practical obstacles meant here? “Our current knowledge on biological factors that guide typical human cortex development is severely limited by practical obstacles»

We would like to refer the reviewer to our common first section of the response letter (subsection “The argument for spatial correlation analyses”) on this issue. In the manuscript, we decided to specifically refer to the core limitations to the study of biological factors underlying human brain development: “Unfortunately, as today’s neuroimaging tools do not suffice to study human *cellular* neurobiology in detail, we have to rely on scarce human *postmortem* and non-human animal data. Conversely, while structures and processes on the *molecular* level are partly accessible in humans with nuclear imaging, here, the exposure to radioactivity practically forbids application in typical developing children and adolescents, limiting its use to study neurodevelopment.”.

6. Introduction: The motivation and goal of the study appears somewhat generic and scattered within parts of the introduction. There are references to biomarkers and some general statements of psychopathology “.. suffer from a lack of reproducible biomarkers and little clinical translation of neuroimaging» and “this knowledge is necessary to understand atypical neurodevelopment and develop targeted biomarkers and treatments». Is the motivation and goal

based in personalized mental health medicine, and that MRI based cortical thickness in the future can be used as a diagnostic tool at the individual level as well as for personalized mental health treatment? This is of course up to de authors, but it could be an idea to tone down such statements and perhaps focus more on the highly valid motivation from the abstract (and end of intro?): “Integrating multilevel brain atlases with normative modeling and population neuroimaging provides a biologically meaningful path to understand typical and atypical brain development in living humans.»

We thank the reviewer for this comment and indeed agree with them in that the focus of our current manuscript lays on understanding brain development rather than on biomarker development (which is a topic we are interested in, but we reckoned that this would go to far for the current, already dense, work). Therefore, we streamlined the Introduction and Discussion in this regard in that we removed the remarks on psychopathology, keeping only the final outlooks on the approach’s potential for individual biomarker development, which could be addressed in follow-up studies.

7. Results 2.1. It is stated that “we report on how these multilevel neurobiological markers colocalize and explain CT change patterns between 5 and 90 years of age». Similarly, to my previous comment, is there not an issue with applying atlases based on adult biology on child and adolescent morphology?

We thank the reviewer for this comment and would like to refer them to our thorough discussion of this issue at the beginning of this response letter.

8. Results 2.1 I struggled to find what the 148-region division was based on. I first searched here “(CT; Fig. 1B; Text S1.2; age distribution: Fig. S5; CT trajectories: Fig. S6A and Animation S1)”, but in the end found a reference to Destrieux within the SI Figure 1 legend, and also later on page 18 under Methods but not stated clearly. Why reduce cortical thickness data based on the Destrieux atlas? This atlas is not based on any of the main candidate mechanisms underlying cortical thickness but instead on larger folding patterns and some anatomical landmarks. As the logic of the study is to use sophisticated cortical divisions, this is not clear to me. A vertex-wise approach would preserve granularity, but then one could not use the open access cortical thickness normative models which I believe are only available for Destrieux and Desikan. I would state and discuss this reduction choice clearly within the text.

Please refer to the response to the question below in which we answer the two comments on parcellation choice jointly.

9. Results 2.1 Similarly, all the cellular and molecular atlases were first reduced to factors and then mapped to Destrieux if I understand correctly. I do not know the initial granularity or spatial distribution of the multilevel atlases, but could reducing them to 74 “folding based” ROIs per hemisphere be problematic? Great with some clarification here. Also, could this mean that your findings are only relevant for studies that use the Destrieux division? I believe both, ROI-based Desikan- or even vertex-wise approaches are more common than using Destrieux.

We agree with the reviewer on the high relevance of the parcellation choice. Indeed, the “space” that spatial colocalizations are performed in is a topic we are very interested in, beyond the current work. As the reviewer correctly points out, the current analyses are all based on the “Destrieux” parcellation with 74 parcels per hemisphere, which was set by the normative model. We would like to note that nearly all prior developmental spatial colocalization studies used the “Desikan-Killiany” parcellation with only 34 parcels per hemisphere, which we explicitly avoided. Based on our experience, a parcel number of about 100–200 is sufficient for spatial colocalization analyses – we are, however, not aware of any systematic investigation of this issue. We agree with the reviewer that, for example, a histoarchitectonically defined parcellation would have been a more reasonable choice (although – to our knowledge – to this day, a full-cortex high-resolution histoarchitectonical parcellation does not exist). We did not run our analyses on a vertex level as this would have overestimated the spatial resolution of the PET data and, much more so, of the gene expression data, here even requiring actively interpolating data. To our knowledge, indeed, most spatial colocalization studies do not work on the vertex level, especially not when working with Allen Brain Atlas gene expression data. Lastly, the factor analyses were computed after parcellation, on the parcel values, across the neurobiological maps. We incorporated this information more explicitly in the manuscript by naming the Destrieux parcellation in the Results (sections 2.1 and 2.2, p. 3–4) and highlighting in the Discussion the need to further explore the relevance of the parcellation choice in future studies (second paragraph, p. 11, “**Generally, although colocalization patterns were similar when evaluated using a coarser anatomical cortex parcellation, we recommend future studies to explore finer divisions based on cortical cytoarchitecture, which could potentially reveal more detailed associations.**”).

To account for the reviewer’s concerns about whether our results apply to the more-often used Desikan-Killiany parcellation, we added a supplementary analysis (described in detail in Suppl. Text 1.3.2), in which we first map both the modeled CT and the neurobiological marker data from Destrieux to Desikan parcels by projecting the data to the fsaverage surface and then re-parcellating them. We then reran the first set of regression analyses. We consider this a sensible approach in a sensitivity analysis setting as the Desikan-Killiany parcellation has a considerably lower number of parcels, likely leading to a case where multiple Destrieux parcels will just be averaged into one Desikan-Killiany parcel. Fig. S11A and B show the Desikan-Killiany-mapped data, which clearly mirrors the patterns of the Destrieux data. Results (Fig S11C and D) show patterns close to the main results but with considerably higher explained variance, notable for both the observed data (colored lines) and the null data (gray shades). This points to overparameterization of the regression models due to the low number of “observations” (i.e., parcels). As noted above, we consider this an important topic to be evaluated in future methodological studies.

10. 2.1. Mapping neurobiological markers to cortical development: Is the cortical microstructure atlas incorrectly referenced? nr 29 (Hansen, J. Y. et al. Mapping neurotransmitter systems to the structural and functional organization of the human neocortex) Is this not the multimodal Glasser atlas? If so, I also believe the correct terms are “T1w/T2w” specifying that this is based on weighted imaging and not (T1/T2) i.e. quantitative relaxometry.

We thank the reviewer for this detailed observation. Indeed, this reference was an oversight from the first draft of this manuscript, written when only Hansen et al. (2022)'s paper was out. It is indeed the T1w/T2w atlas from the neuromaps toolbox, which cites Glasser et al. (2016) as the source. Both the references and the naming were corrected.

11. 2.1. Mapping neurobiological markers to cortical development: As the Rutherford normative models are based on cross-sectional data I would not call these trajectories cortical “development” of even “change”, as they are based on cortical differences at different ages or? I would be consistent in the terminology and only use development and change when talking about longitudinal data like ABCD/Imagen.

We understand the concerns of the reviewer on this issue. It is correct that the normative model has been estimated on cross-sectional data, which we now explicitly mention in the results section (section 2.2, “CT trajectories for 148 Destrieux regions were derived from a normative

model of CT development³ estimated from cross-sectional data of over 58,000 subjects (from here on referred to as “modeled CT” [...])). As noted in response to another comment above, we now use the term “modeled CT (change)” whenever we write about the data obtained from normative model predictions. While we agree with the reviewer that estimating longitudinal changes from cross-sectional data often results in less precise estimations, i.e. due to cohort or other confounding effects, we would argue that it is justified to talk about “change” in this instance as we are calculating the age-related CT change predicted by a model developed for modeling of lifespan/longitudinal “changes” in cortical development.

12. Discussion: It was slightly unclear to me whether your findings match the previous main candidate mechanisms for cortical development. To my understanding re-organization of dendritic arbour, cortical myelinization, and even synaptic pruning (although controversial as MRI studies cannot be solely explained by small-scale changes at the synapse level) are candidates. Your discussion instead highlight microglia and thus immune responses and dopamine receptor activity is that correct? When I read the discussion, it currently appears as if all your results fit previous literature with no discrepancies discussed. Also, could the relation to the dopamine system be considered controversial just in regard to the mm scale of MRI?

We thank the reviewer for this insightful comment, agreeing with their standpoint. In response, we rewrote the first Introduction paragraph to explicitly introduce and name neuronal remodeling (reorganization of dendritic arbor, synaptic pruning) and cortical myelination as the processes mainly discussed as underlying macroscale CT changes. In line with this and other comments, we also edited the Discussion, now providing a more balanced allocation of our findings in the literature and especially also discussing existence and treatment of non-converging findings (first and second Discussion paragraphs). Regarding the last point, we would like to also point to one of the preceding comments on the hierarchical introduction of potential mechanisms shaping cortex macrostructure. The reviewer is right in that we would not assume that a change in cortex macrostructure would be directly caused by neurotransmitter receptor density measured with PET. Rather, one could hypothesize a mechanism for which the dopaminergic receptor distribution serves as a proxy, e.g., as referenced in the paper, a regulatory role of dopamine/dopaminergic receptors in cellular reorganization, or possibly also a developmental change in reward or attention networks. However, we emphasize that such hypotheses are relatively far-fetched, given the current work, which we think should rather serve as a starting point for further investigation

13. Discussion: It is stated that “As promising candidates for clinical translation, we identify the dopaminergic system and microglial cell populations for early development”. Would one not have to apply the normative models to clinical populations like in the Rutherford paper, in order to probe clinical relevance? It is unclear to me why non-typical age trajectories should relate to mental health any more than to any other variable that shows stronger associations to imaging and cortical thickness specifically. Like a previous comment, I find the link to clinical utility inconsistent, and in my opinion not necessary for the relevance of the paper.

We agree with the reviewer on this issue, and as described in the above comment to which the reviewer also refers, we mostly dropped these clinical connections from the paper. Only in the last Discussion sections, we provide outlooks to potential clinical follow-up work.

14. Methods: It is stated that “Given that the main goal of the current analysis was to establish the feasibility of capturing associations between CT development and multilevel neurobiological markers on the individual level, clarification of the sources of .. site-effects will be a task for future investigations. When using multi-site initiatives like ABCD (21 sites, 29 scanners), harmonizing or somehow minimizing the effects of scanner is basic convention. It is not clear why the authors find this a task for the future as the large effects of scanner is already widely documented also specifically within ABCD. The normative models used within the paper attempt to minimize effects of scanner, should the same not pertain to the target longitudinal data? Please explain why scanner bias is not relevant for your analyses or add it as a limitation.

Indeed, given the scope of the current paper, we changed our strategy to multi-site harmonization in the ABCD/IMAGEN data significantly: Instead of calculating all “observed single-subject” analyses on the model-adapted CT (change) data, we kept the ABCD and IMAGEN data completely independent from the model. This also serves another obvious purpose, which is to exclude any potential bias that the model-adaptation procedure could have on the single-subject data (i.e., to exclude that the adaptation introduced the “CT change explained”-effects into the ABCD/IMAGEN data). In the revised version, all analyses based on observed single-subject ABCD/IMAGEN cortical thickness use data that was harmonized using ComBat-GAM as a validated state-of-the-art method [NeuroHarmonize (Pomponio et al., 2020), modeling age as a non-linear covariate]. The procedure is described in the Methods (4.3.3). Notably, harmonization was performed on the cross-sectional data before calculating change metrics. In addition we still ran the model adaptation procedure, however only to provide comparisons between model-

predicted and observed “CT change explained” effects (e.g., Figs. S19–S24) and to visualize the single-subject values together with model predictions (e.g., Animation S1). The systematic comparison of CT and CT change metrics between sites showed that the ComBat-GAM harmonization removed site effects between timepoint-wise CT for ABCD but only reduced site effects for IMAGEN (compared to non-harmonized data). For CT change, site effects were also reduced but apparent in both datasets (Tables S3 and S4). As in the original manuscript, the level to which CT change could be explained by neurobiological markers also differed clearly between sites. However, with ComBat harmonization, explained CT change in the single-subject datasets increased slightly (by about 1%, updated results in Fig. 6). Using the updated site-harmonization method, we aimed to make sure that the observed site-effects were not due to potentially insufficient site-correction by the normative model adaptation procedure. However, generally, we would like to point out that we consider site effects not to be of high relevance to our results as we report within-subjects estimates and do not, for example, compare data across groups with different site compositions (where “site” would be a very relevant potential confound!). It would go beyond the scope of the current work to dive deep into the potential sources of the observed site variation, which might not only be due to between-scanner variance but potentially also due to other biological or demographic factors, leading to variance in cortex developmental trajectories (Suppl. Text S1.4.3). Finally, to accommodate the request of the reviewer, we included further exploration of site effects in our final paragraph on necessary further steps to explore potential clinical value of the “developmental spatial colocalization” framework (fourth Discussion paragraph, p. 13).

15. Methods It is stated that site effects were estimated in «healthy subsamples» of both dataset’s baseline data (n = 20 per site, 50% female) how was healthy defined?

We would like to refer to the above comment, noting that, with new harmonization strategy, the main analyses of single-subject ABCD/IMAGEN data are not anymore based on the model-adapted data, but instead site effects were estimated on the whole datasets and “removed” from the whole datasets (separately from ABCD and IMAGEN) using ComBat-GAM. Due to word limit constraints, we now provide the requested information in the supplements (Text S1.4.1): “For this purpose, “healthy” was defined as: no psychiatric diagnoses according to KSADS-parent (ABCD) or DAWBA (IMAGEN); no medical diagnoses according to “abcd_mx01.txt” (ABCD-only), and no history of traumatic brain injury (ABCD-only).”

Reviewer #3:

I previously reviewed this paper for Nature. I have not retained a copy of the version submitted to Nature, and the authors have not provided a rebuttal letter detailing their response to reviewer comments on that submission, so it is difficult to be sure if/how this version has been revised compared to the version previously reviewed for Nature. I think some of my points have been minimally addressed in Discussion, however there don't appear to have been many, if any, substantive changes. Most of my prior comments therefore still stand, as copied below:

"This paper reports the results of an extensive set of analyses, using multiple large prior datasets, conducted by an expert and experienced team using careful and generally well-described methods. Overall the text and figures are composed to a very high standard.

The study investigates spatial colocation of cortical thickness (CT) with multiple molecular imaging maps (referred to as nuclear imaging) and maps of neurotransmitter density and cellular density derived from postmortem data. The basic concept of colocalising MRI phenotypes with genomic, molecular and cellular reference atlases in an effort to understand more about the neurobiological substrates of MRI phenotypes is not novel. This strategy has been widely used in recent years and it has generated interesting results, as in this paper, but it also has well-known limitations. It is clear that the authors are aware of all this. However, I am not sure how decisively this work advances the field beyond what has already been done, or tackles some of the fundamental limitations of spatial colocation analysis for mechanistic interpretation of MRI phenotypes.

The MRI CT maps are derived from normative modelling of ~50,000 scans as previously reported by Rutherford *et al.*, and from the ABCD and IMAGEN datasets, to represent the cortical distribution of thickness as a function of age. Each age-specific CT map is then assessed for spatial co-location with factors derived from 49 prior atlases (Supp Info Table 1). 21 of the atlases are cell-specific gene expression patterns previously derived from the Allen Brain Atlas. Also included is a T1/T2 myelin map from the Human Connectome Project. The remaining 27 atlases are derived from PET data using various transmitter-specific and other radiotracers. The molecular (imaging) and cellular (post mortem) atlases were then separately subject to factor analysis, resulting in 10 molecular and 10 cellular factors + the T1/T2 map = 21 so-called "multi-level brain systems". Each of the 21 factors was then tested for spatial colocation with the CT data cross-sectionally and with

CT change scores for 5 year epochs by a sliding window analysis in the age range 5-90 years. I have a number of comments, questions about this approach.

We thank the reviewer for their positive comments on our methodology and reporting quality, and we are grateful for them taking the time to review our manuscript a second time.

Indeed, the first version of this manuscript was submitted to Nature nearly a year ago, in May 2023. After receiving the rejection by Nature, the manuscript was revised and submitted to another journal prior to the submission to Nature Communications. Because of this intermediate process, we did not provide a point-by-point response to the very initial reviews. We apologize for this inconvenience. We respond to each of the reviewer’s comments below, separating what was changed in response to the “old” reviews commissioned by Nature and in response to the current reviews.

Generally, it is worth noting that the abbreviation “MLBS” that the reviewer uses refers to the term “multilevel brain system”, which we used in the version of the manuscript submitted to Nature. In response to the initial reviews, we replaced the term by “multilevel neurobiological marker”, avoiding the term “system”, which was not straight-forward to apply to, e.g., a cell population. In the now-revised version of the manuscript, we decided to use only “neurobiological marker” for clarity and simplicity.

1. The PET atlases are quite diverse in terms of age (mean age ranging from 22-68 years across atlases) and sample size (N ranging from 6-204). The ABA atlases are derived from N=6 with mean age 42.5 year. None of the reference atlases provides data over the same age range as the CT trajectories. The implications of this mis-match are briefly mentioned (as “noise”, in the last para of discussion) but not properly addressed. Do the authors assume that the cortical distribution of, say, the dopamine transporter (DAT), is unchanged over the life-cycle, and therefore that the available DAT map (mean age 61 years) is a reliable indicator of DAT distribution in adolescence? This assumption seems unlikely to be true, either for DAT or in general across all the reference atlases; but without it how we can make any mechanistic interpretation of colocation between developmentally varying CT measurements and fixed-in-time reference atlases? As far as I can see, the authors’ principal defence on this point is that some of their results are consistent with what is already known about the developmental dynamics of some molecules or cells in the brain. For example, in the Discussion, they note that the strong colocation of the dopamine-based factor with CT change in adolescence is consistent with prior data showing that D1 receptor density peaks

during adolescence. Fig 6 descriptively summarises the context for some of the results reported in this paper compared to prior reports of developmental change in other molecules or cells. Personally, I didn't find this very convincing. I understand that there are limitations to the data available for use as reference atlases for lifecycle development – but despite identifying this as a problematic aspect of prior studies using the ABA dataset (in Introduction), and their brief comments in Discussion, the authors could do more to convince the reader that spatial co-location of two maps measured at very different points in the lifecycle is mechanistically relevant (rather than descriptively interesting) for deeper understanding of brain developmental processes. I suggest this is particularly important because the analysis is correlational not causal, and the causal mechanistic significance of co-located “multi-level brain systems” and MRI maps would be debatable if the MLBS and MRI data were measured in demographically matched cohorts, or even from combined PET/MRI data in the same cohort, e.g., do dopamine receptor changes in adolescence drive CT changes, or vice versa?

We thank the reviewer for the very detailed comments on this issue, which assisted us significantly in revising the manuscript. Most importantly, the major change made to the manuscript after receiving the feedback by Nature, was the inclusion of validation analyses using developmental gene expression data (Results 2.8, Methods 4.6, Figs. 1, 5, and S17). Furthermore, assisting us as an expert in the analysis of this data, Dr. Casey Paquola was included as co-author. These analyses were performed given the reviewer's focus on providing reassurance for our results. However, given the overall feedback from the current round of reviews, we put the focus of this revision on the topic of the mechanistic assumption underlying our methodological approach that the reviewer raised. For this, we would like to refer to our detailed exploration of this issue in the general first part of this response letter and hope to have responded sufficiently to the reviewer's comments both in our responses as well as in the revised manuscript.

2. There is insufficient detail provided about some aspects of the factor analysis used to generate “multi-level brain systems”. The following points were not clear to me: How were the reference atlases differentially weighted on each factor? How important was each of the 10 cell (or molecular) factors in terms of the proportion of total variance in the “raw” atlases that they accounted for? What was the extent of correlation between MLBS's derived from non-orthogonal factor analysis? Without these data it is very difficult for a reader to make up their own mind about interpretation of the results of correlating these derived factors with CT maps. The rationale for

progressively stepping down from 21 MLBS's to 9 and then 6 and then focusing on a few “raw” PET atlases, over the course of the results, could also have been more clearly signposted in advance or justified.

We thank the reviewer for raising this issue. Indeed, the reporting of the relationships between the original and dimensionality-reduced neurobiological markers, as well as of the factor analysis details did not change significantly between the manuscript versions submitted to Nature and Nature communications. We decided to keep the reporting as all information that the reviewer requested was already included in supplementary text and figures. However, in the revised version, we now provide the following information in the first Results subsection: “**Intercorrelation arising from spatial patterns shared between atlases on either cellular or molecular levels (Fig. S3A) was reduced by factor analyses applied independently to the cellular and molecular marker sets after parcellation of the individual maps. For each marker set, all unrotated factors that explained at least 1% of the set's variance were retained, resulting in 10 “factor-level” nuclear imaging maps (*ni1–10*) and 10 gene expression cell marker maps (*ce1–10*). After promax rotation, these factors explained 90.9% and 86.9% of each marker set's variance, respectively (Fig. S3B and C). We chose the liberal factor-number criterion to balance retaining as much of the spatial information in the neurobiological markers as possible with reducing marker multicollinearity in the following multivariate analyses. Factor solutions were successfully validated against permuted brain maps (Text S1.2.4) and factors were named based on the most closely related original atlases (Fig. S3D and E).**”. Additionally, we provide a description in the main Methods (4.3.1) and a description of the permutation test in the supplements (Text S1.2.4). We provide factor loadings as well as correlation matrices for (i) the original maps among each other, (ii) the original maps with each factor map, and (iii) the factor maps among each other in Fig. S3. Indeed, the permutation test was added in response to the initial reviews via Nature to demonstrate that the estimated factor-solutions explained significantly more variance in the original data than “random” solutions (estimated on permuted neurobiological marker maps).

In response to the second comment, we now describe our hierarchical workflow (from all dimensionality-reduced markers, to selected dimensionality-reduced markers, to original markers that loaded strongly on the selected dimensionality-reduced markers) more explicitly in Results section 2.2 to inform the reader beforehand about what to expect.

3. It is commendable that the authors endeavour to generalise results derived from the Rutherford et al MRI dataset to ABCD and IMAGEN datasets including longitudinal data. It is reported that the MLBS co-location results reported on the basis of the Rutherford dataset are somewhat replicated in the ABCD and IMAGEN datasets (R^2 25-56%). Since the MLBS's are unchanged for the Rutherford and ABCD/IMAGEN analyses, presumably this degree of correspondence simply reflects the correlation between Rutherford and ABCD/IMAGEN CT maps? Likewise, the high degree of variability in MLBS-coupling noted at an individual level presumably reflects the typicality or atypicality of an individual's CT phenotype compared to the developmental norms published by Rutherford? I apologise if I have missed something in the rather dense description of these analyses but showing that datasets A and B (Rutherford and ABCD/IMAGEN) are both correlated with fixed point of reference C (MLBS) surely only tells us that A and B are correlated? This would not be surprising, especially since the Rutherford dataset includes (baseline, cross-sectional) data from the ABCD cohort, i.e., A and B are not entirely independent of each other by construction. I would welcome greater clarity concerning the relationships between the MRI phenotypes (A and B), and the extent to which variability in individual MLBS-coupling reflects individual (a)typicality of MRI phenotypes compared to age-appropriate norms. As it stands, I found it difficult to see how the analysis of ABCD/IMAGEN longitudinal data, in aggregate or individually, added materially to the robustness or biological interpretation of the principal results derived from the Rutherford dataset.

We agree with the reviewer on their interpretation of the spatial intercorrelation between CT (change) distributions in the modeled (Rutherford), ABCD, and IMAGEN data. However, we think that there might be a misunderstanding on our motivation for the individual-level analyses: The normative model is trained on cross-sectional data and, by its nature, models CT trajectories as smooth curves. We use predictions from this model to calculate “modeled” timepoint-to-timepoint CT changes, assuming that these actually show average (median) cortex development patterns as would be observed in longitudinal data, even though the model was trained on cross-sectional data. We then use the actual longitudinal data (i.e., ABCD and IMAGEN) to show that this is indeed the case on the level of “explained variance of CT change patterns”. As can be seen in the analyses “predicting” cohort-average CT changes across the different timespans, the changes between predictions by the model indeed seem to correspond to the actual changes in CT observed in longitudinal data (e.g., Fig. 6A, top and middle row). As the reviewer points out, on the

individual-subject level, we find high interindividual variance, although the overall pattern of explained CT changes seems to come through, nevertheless. We agree with the reviewer that the correspondences between normative model predicted data and actual longitudinal data in two separate cohorts stem from the fact the spatial patterns of CT changes are shared between these data sources. This might sound rather trivial, but given the cross-sectional training data of the normative model and a motivation to assess how the “CT change prediction” frameworks performs on the individual level, in our view, justified the independent longitudinal analyses. Therefore, we would indeed argue that they add to the robustness of our results, but we agree with the reviewer in that they do not necessarily add to the biological interpretation of the model-based results at the current stage. The latter could be achieved, however, with further follow-up projects, for example, evaluating epi-(genetic) or peripheral-physiological sources of the observed interindividual variation, which we think would go beyond the scope of the current work. To accommodate these thoughts in the manuscript, we changed the introduction of the individual-level analyses in the Results, highlighting that the exploration of sources of interindividual variability was the core motivation (section 2.9, “However, a sole focus on modeled population CT change, i.e., median predictions from the normative model, does not allow for inferences about individual-level neurodevelopment, which is the mandatory prerequisite for exploring potential sources of interindividual variability.”), and specifically discussed the discrepancy between cohort- and individual-level explained CT change estimates in the final Discussion paragraph (fourth paragraph, “As evidenced by our comparative analyses between “CT change predictions” by the normative models and cohort-average CT changes as observed especially in the independent IMAGEN sample, normative models are capable of predicting population-level development, even when estimated only on cross-sectional data. Going beyond this group-level discovery approach, we demonstrate the feasibility of developmental spatial colocalization analyses in single subjects by mapping individual-level brain development to specific neurobiological markers. However, the strong variation in colocalization estimates observed on the individual level warrants further research into normative modeling of longitudinal data on the one side and potential sources of interindividual variability on the other.”), while highlighting potential future study directions. Last, we are aware that we could have shown the correspondence of CT change patterns between normative model, IMAGEN, and ABCD data on the “CT change” level rather than on the “CT change explained” level. Indeed, we provide overview plots for such relationships in Figs. S19 and S20. However, as the CT change explained is our – to some extent – biologically interpretable main

output metric and, given our core motivation of providing an individual-level metric with potential biological meaning to be evaluated further from physiological and clinical standpoints, we argue that the focus on explained CT change in the longitudinal analyses was the most reasonable route to take.

4. The Discussion suggests that these methods and results can be used to validate new treatment targets, and/or to understand the molecular or cellular basis of atypical MRI phenotypes in neurodevelopmental disorders. However, there are no data reported on disorders and the only "targets" suggested by these results are "dopaminergic and microglial systems for early development as well as the cholinergic system in context of pathological aging". This level of target validation is too broad-brush for practical purposes, e.g., drug development, and I'm not convinced it tells us anything we didn't already know. Likewise, I think the claim (p15) that "spatial colocalization approaches can facilitate discovery of physiological mechanisms underlying specific conditions" is not really nailed down by the data that have been reported, due to the limitations of spatial correlational analysis for clarifying mechanistic relationships between correlated maps."

We thank the reviewer for this comment, which was also raised by the other reviewers. In our thorough revision of the Introduction and Discussion, the statements the reviewer refers to were indeed removed or significantly altered. As mentioned in responses to the other reviewers' comments, we have now removed the references to neurodevelopmental or neurodegenerative disorders to streamline our reporting keeping the main focus on non-pathological cortex development. Therefore, the first sentence ("dopaminergic and microglial systems [...]") was dropped from the paper. The statement on drug development was likewise removed. The last statement ("[...] specific conditions") was altered to "Our results indicate that if used as a reference for typical brain development, combining normative models of brain regional features with spatial colocalization approaches could facilitate discovery of physiological mechanisms underlying specific developmental patterns.", which we reckon stays closer to our analyses and findings. In the Discussion, we explore the clinical viewpoint from two perspectives that both highlight potential future research avenues: First, clinical data could be used to provide validation for the spatial colocalization markers (third Discussion paragraph, p. 12) in terms of showing that spatial colocalization estimates are altered in patterns expected for a specific disorder (e.g., the D1/D2 colocalization in ADHD or psychosis). Second, one could explore clinical features as sources of interindividual variation in the spatial colocalization metrics (last Discussion paragraph, p. 13).

However, as this work is already very extensive, we believe that such endeavors belong in separate projects rather than being integrated into the current one.

Reviewer #1 (Remarks to the Author):

The authors have made a significant revision. I have only a few minor comments and suggestions.

1) The authors asserted the development of the human cortex throughout the manuscript, yet their focus was solely on cortical thickness. It is evident that different cortical morphological features, such as cortical thickness, surface area, and volume, exhibit distinct growth patterns and are likely to demonstrate varying neurobiological associations. Consequently, I would suggest the authors to rephrase their claims, with a primary focus on cortical thickness. It would also be valuable to include a brief discussion on this topic.

2) The authors performed spatial correlation analyses between developmental cortical thickness and adult neurobiological markers. While the authors have added descriptions of scientific reasonability in the response letter, the discussion of limitations in the main text is insufficient. I would suggest including a comprehensive discussion of the major limitation.

Reviewer #2 (Remarks to the Author):

The authors have successfully addressed all my concerns, and I have no further comments. (Also, I sincerely apologize for the delayed response due to my maternity leave).

Reviewer #3 (Remarks to the Author):

Thanks to the authors for their very thorough and constructive response to the issues I raised - I am now happy to recommend the paper for publication

Dear Editorial Team, dear Reviewers,

We thank you again for the time and effort you have put into evaluating our revised manuscript. Below you find our responses to the reviewers' comments. We hope that these sufficiently address the remaining points raised by Reviewer #1. All original reviewer comments are printed in **blue font**, citations from the revised manuscript are printed in **red**. We thank you for considering our manuscript for publication!

Sincerely,

Leon D. Lotter and Juergen Dukart, on behalf of the authors

Reviewer #1:

The authors have made a significant revision. I have only a few minor comments and suggestions.

We thank the reviewer for acknowledging our detailed revision. We hope that our responses below and adjustments to the manuscript sufficiently address the reviewer's remaining concerns.

1) The authors asserted the development of the human cortex throughout the manuscript, yet their focus was solely on cortical thickness. It is evident that different cortical morphological features, such as cortical thickness, surface area, and volume, exhibit distinct growth patterns and are likely to demonstrate varying neurobiological associations. Consequently, I would suggest the authors to rephrase their claims, with a primary focus on cortical thickness. It would also be valuable to include a brief discussion on this topic.

We agree with the reviewer on our perhaps imprecise language in parts of the manuscript. We adjusted the text while keeping a close eye on what level (i.e., the broader “cortex” or “cortical” vs., more specifically, “cortical thickness”) we referred to exactly in each statement. In the revised version, we use the broader “(development of the) cortex” only in introductory sentences and in references to prior literature or our general approach as appropriate. When we talk about our specific methodology and results, we consistently use “cortical thickness”/ “CT”. We would like to note that we maintained the current title (“Regional patterns of human cortex development spatially overlap with underlying neurobiology”) as it was specifically suggested by the editor.

Finally, we added a statement on the topic matter to the last discussion paragraph, citing the well-known reports on diverging genetic influences and developmental trajectories of cortical

thickness and area by Wierenga et al. and Panizzon et al.: “Notably, here we focused exclusively on CT. Considering the diverging genetic influences and developmental trajectories of other brain-morphological features^{81,82}, those might show different neurobiological associations.”

2) The authors performed spatial correlation analyses between developmental cortical thickness and adult neurobiological markers. While the authors have added descriptions of scientific reasonability in the response letter, the discussion of limitations in the main text is insufficient. I would suggest including a comprehensive discussion of the major limitation.

We thank reviewer #1 for reemphasizing this central point of our work. However, while the reviewer only points out our “descriptions of scientific in the last response letter”, we would like to first highlight the thorough revisions made to the manuscript in this regard:

We introduce our approach noting the difficulties of directly measuring biological processes or properties of the human brain in vivo (second Introduction paragraph, “Unfortunately, as today’s neuroimaging tools do not suffice to study human cellular neurobiology in detail, we have to rely on scarce human postmortem and non-human animal data. Conversely, while structures and processes on the molecular level are partly accessible in humans with nuclear imaging, here, the exposure to radioactivity practically forbids application in typical developing children and adolescents, limiting its use to study neurodevelopment.”). We then provide a short form of the mechanistic argument for our spatial colocalization approach, which is what the reviewer might refer to above (Introduction, second paragraph, starting with “Neural cell populations and molecular-level tissue structures and processes [...]”). The complete version of this argumentation (closely following the response letter) was added to the supplementary materials. Notably, our argument is specifically based on correlating developmental MRI data with adult-derived “normative” atlases of neurobiological properties! However, we acknowledge other interpretations of our method and results, as noted below. At the end of our argumentation in the main Introduction, we note the possibility of “third” processes influencing the correlation between CT change and neurobiological markers, limiting specificity of interpretation (“Notably, it is conceivable that a third process could influence both, X and CT, leading to a correlation between X and CT changes that is non-causal but still implies a neurobiological mechanism influencing both.”). We close with listing further limitations, including the limited causal inferences (“We emphasize that, despite causal assumptions being made on the conceptual side (see above), neither prior nor the current spatial correlation study can actually prove causal relationships between an

MRI-observed change pattern and tested neurobiological markers. Relatedly, the specificity of spatial associations is inherently limited by the spatial resolution and noise associated with both correlated components.”).

In the largely rewritten Discussion section, we list several limitations of our approach, however, not in a dedicated Limitations section but at the contextually sensible places in the Discussion (e.g., second Discussion paragraph, starting from “**Interpretation of our results is furthermore complicated by [...]**”). Here, we note several limitations of both the adult-derived neurobiological atlases as well as the cross-sectional lifespan CT models. We then go on to discuss what would be necessary to infer causal relationships from spatial colocalization analyses (third Discussion paragraph) as well as what is required for a potential clinical translation (last Discussion paragraph).

Given these substantial improvements made to our manuscript in terms of a future-oriented discussion of limitations of the present study as well as spatial colocalization approaches in general, which already maximally increased the length of our manuscript, we suggest a rather concise way of handling the reviewer’s request: With adding the following sentence the very end of the manuscript’s Discussion, “**Closing, it is important to reiterate that the central assumption of the spatial colocalization approach used here is that the spatial topology of adult neurobiology is reflected in neurodevelopmental changes in MRI-based CT estimates. Given the lack of non-invasive tools to study developmental neurobiology in vivo, our study provides evidence, albeit indirect, for such associations.**”, we hope to lastingly remind the reader of this central limitation (as raised by the reviewer), however in light of the necessity of the current work.

Reviewer #2:

The authors have successfully addressed all my concerns, and I have no further comments.

(Also, I sincerely apologize for the delayed response due to my maternity leave).

We are happy that we were able to address all raised concerns and thank Reviewer #2 for their positive evaluation of our manuscript!

Reviewer #3:

Thanks to the authors for their very thorough and constructive response to the issues I raised - I am now happy to recommend the paper for publication.

We thank Reviewer #3 for their favorable response to our revision!